# Polymer Membranes Sonocoated and Electrosprayed with Nano-Hydroxyapatite for Periodontal Tissues Regeneration

**DOI:** 10.3390/nano9111625

**Published:** 2019-11-15

**Authors:** Julia Higuchi, Giuseppino Fortunato, Bartosz Woźniak, Agnieszka Chodara, Sebastian Domaschke, Sylwia Męczyńska-Wielgosz, Marcin Kruszewski, Alex Dommann, Witold Łojkowski

**Affiliations:** 1Laboratory of Nanostructures, Institute of High Pressure Physics, Polish Academy of Sciences, 01142 Warsaw, Poland; b.wozniak@labnano.pl (B.W.); a.chodara@labnano.pl (A.C.); w.lojkowski@labnano.pl (W.Ł.); 2Faculty of Materials Science and Engineering, Warsaw University of Technology, 02507 Warsaw, Poland; 3Laboratory for Biomimetic Membranes and Textiles, Empa Swiss Federal Laboratories for Materials Science and Technology, 9014 St. Gallen, Switzerland; 4Experimental Continuum Mechanics, Empa Swiss Federal Laboratories for Materials Science and Technology, 8600 Dübendorf, Switzerland; sebastian.domaschke@empa.ch; 5Department of Mechanical and Process Engineering, Institute for Mechanical Systems, ETH Zürich, 8092 Zürich, Switzerland; 6Centre for Radiobiology and Biological Dosimetry, Institute of Nuclear Chemistry and Technology, 03195 Warsaw, Poland; sylwia.meczynska@gmail.com; 7Department of Molecular Biology and Translational Research, Institute of Rural Health, 20090 Lublin, Poland; m.kruszewski@ichtj.waw.pl; 8Department Materials meet Life, Empa Swiss Federal Laboratories for Materials Science and Technology, 9014 St. Gallen, Switzerland; Alex.Dommann@empa.ch

**Keywords:** electrospinning, sonocoating, electrospraying, periodontal regeneration, GTR/GBR membranes

## Abstract

Diseases of periodontal tissues are a considerable clinical problem, connected with inflammatory processes and bone loss. The healing process often requires reconstruction of lost bone in the periodontal area. For that purpose, various membranes are used to prevent ingrowth of epithelium in the tissue defect and enhance bone regeneration. Currently-used membranes are mainly non-resorbable or are derived from animal tissues. Thus, there is an urgent need for non-animal-derived bioresorbable membranes with tuned resorption rates and porosity optimized for the circulation of body nutrients. We demonstrate membranes produced by the electrospinning of biodegradable polymers (PDLLA/PLGA) coated with nanohydroxyapatite (nHA). The nHA coating was made using two methods: sonocoating and electrospraying of nHA suspensions. In a simulated degradation study, for electrosprayed membranes, short-term calcium release was observed, followed by hydrolytic degradation. Sonocoating produced a well-adhering nHA layer with full coverage of the fibers. The layer slowed the polymer degradation and increased the membrane wettability. Due to gradual release of calcium ions the degradation-associated acidity of the polymer was neutralized. The sonocoated membranes exhibited good cellular metabolic activity responses against MG-63 and BJ cells. The collected results suggest their potential use in Guided Tissue Regeneration (GTR) and Guided Bone Regeneration (GBR) periodontal procedures.

## 1. Introduction

Expanding knowledge about nanomaterials has led in recent years to the advent of nanomedicine and the transformation of medical and dental clinical strategies. Major advances and innovations have been made, especially in the fields of nanomaterials for periodontal regeneration, where the main challenge is to achieve the regeneration of alveolar bone tissue lost due to inflammatory processes [1,2,3]. Bone, a natural nanocomposite, is made of organic compounds (mainly collagen) and inorganic phase-hydroxyapatite nanocrystals. This unique architecture indicates the potential use of synthetic nano-hydroxyapatite in orthopedics and dentistry applications. The purpose of this study was to exploit opportunities given by combining the bone regeneration potential of nanohydroxyapatite (nHA) with the good mechanical properties and bioresorbablility of micrometric, electrospun fibers to create promising materials for periodontal bone tissue regeneration. 

In dentistry, diseases of periodontal tissues are a major clinical problem, which affect up to 70% of the global population: adolescents, adults, and elderly individuals. Several factors such as smoking, insufficient oral hygiene, diseases, age, medications, and stress increase the risk of periodontal diseases. When the periodontal disease progresses, periodontium tissues are damaged, leading to tooth loss. The ultimate goal of periodontal regeneration treatment is to reduce inflammation and regenerate tissue lost due to the disease. Guided Tissue Regeneration (GTR) and Guided Bone Regeneration (GBR) surgical procedures are the current restoration strategies used to overcome this problem [4,5,6]. Both methods are based on the principle of separating soft tissues from the bone cavity with a membrane. The membrane should prevent soft tissues from ingrowth in the bone defect area and enhance tissue regeneration by guiding cell proliferation on both sides. In the GBR procedure, the membrane protects the bone defect site from the rapidly-growing epithelial cells, as well as fibrous and gingival tissues ingrowth. It improves both the soft tissue regeneration and bone regeneration. In the GTR procedure, the membrane covers the periodontal defects around the root of the tooth, stops in-growth of the cells, and ensures stability of blood clots. Consequently, the wound area and the previously defective tooth root surfaces can be repopulated by cells originating from the intact periodontal ligament [7,8]. One of the key requirements for GTR/GBR membranes is degradability to enable up to 5 months’ healing of the defect [5,6]. The membranes need to exhibit biocompatibility to allow integration to occur with surrounding tissues without causing inflammatory responses, good mechanical and physical properties to allow its placement in vivo, and sufficient stiffness to enable handling during surgery [9,10,11]. Non-biodegradable and biodegradable membranes, such as Gore-tex® (W.L. Gore & Associates, Newark, DE, USA), Guidor membrane (Sunstar Group, Etoy, Switzerland), Vicryl mesh (Ethicon, Cincinnati, OH, USA), and Bio-Gide® (Geistlich Biomaterials, Wolhusen, Switzerland), are commercially available for such periodontal surgery purposes. Non-degradable, expanded polytetrafluoroethylene (e-PTFE) and titanium mesh membranes were frequently used for alveolar bone regeneration. Non-biodegradable membranes need to be removed by an additional surgical operation, which involves additional costs, and may led to inflammation at the operated site. For this reason, various biodegradable polymer membranes have been developed to eliminate the need for revision surgery [12,13,14]. However, current clinical periodontal therapies do not sufficiently promote complete regeneration of periodontal tissues [15,16]. Therefore, new approaches and materials are needed to regenerate the lost periodontal tissue [17,18].

To fabricate fibrous membranes for periodontal tissue regeneration, numerous research groups have explored the electrospinning method [19,20]. Electrospinning has received much attention recently due to the growing interest in nanotechnologies and the properties of unique fibrous materials. It was reported that electrospun membrane scaffolds stimulate positive cell responses, increase proliferation, and provide physical and chemical stimuli to cells [21,22,23]

The aim of the present study was to develop a novel composite membrane with a desirable degradation rate and improved wettability, with a potential to enhance the tissue regeneration process. Polymer blends of Poly (D,L-lactic acid) (PDLLA)/poly(D, L-lactic- co-glycolic acid) (PLGA) with tunable degradation rates were chosen to develop the fibrous membrane by electrospinning. Both polymers have been extensively used in medicine due to their good biocompatibility and biodegradability. Electrospun membranes for load-bearing applications in implantology need to exhibit sufficient mechanical properties to allow placement in defects, avoiding membrane collapse and providing barrier function. PDLLA has a hydrophobic nature and displays a slow degradation rate, i.e., up to 12 months, which limits their use in GTR/GBR [24,25,26]. The degradation rate and hydrophilicity of PDLLA can be regulated by blending it with the faster degrading PLGA [27,28,29,30,31]. However, some polymeric implants during degradation may release acidic by-products and cause localized foreign body reactions, characterized by presence of macrophages and intracellular remnants of the polymer, which may cause an immune response [32]. Further, the surface properties of the polymer fibers obtained by electrospinning are not optimized for tissue engineering applications. A strategy to reduce the aforementioned shortcomings of polymers in GTR/GTB applications is to modify their properties by adding nHA, which is chemically similar to the mineral phase found in natural bone, and has been widely used in bone tissue engineering due to its bioactivity and osteoconductivity. In that respect, there are two main approaches: coating and producing nanocomposites. Zou et al. [33] reported that that presence of hydroxyapatite in polymeric matrices leads to high water absorption and neutralization of the acidic environment [34]. It also slowed down autocatalysis during polymer biodegradation [35]. Many studies have shown that nHA promotes osteoblast adhesion, differentiation, and proliferation, resulting in faster osteointegration and enhanced formation of new bone tissue [36,37,38,39,40,41]. Ceramic particles embedded in the polymer matrix surface would improve the biocompatibility and hard tissue integration due to higher wettability compared to the more hydrophobic polymer. It was reported by Lao et al. that the addition of hydroxyapatite to a PLGA electrospinning solution increases the proliferation of mouse pre-osteoblasts (MC3T3-E1) cultured on fibrous material [42]. 

An alternative approach is coating the fibers with nanometric hydroxyapatite. The nano-particle coating should increase the high specific surface area and wettability, and promote protein adsorption, thereby enhancing the spread of bone cells. Additionally, surfaces rough in the nano-scale have been shown to favor bone tissue integration, which is important for the design of bone-contacting implant materials [43,44,45]. A known approach for the deposition of hydroxyapatite on electrospun fibers is electrospraying [46]. Gupta et al. reported that the electrospraying of HA nanoparticles, in combination with the electrospinning of fibers, offered favorable surface topographies and an osteophilic environment for the attachment and growth of human fetal osteoblastic hFOB cells [47]. Ramier et al. fabricated the tissue engineering scaffold with high surface roughness by electrospraying nHA on the Poly(3-hydroxybutyrate) (PHB) electrospun fibers [48]. Garcia Garcia et al. reported the fabrication of a 3D multilayer scaffold with HA particles electrosprayed on the fibers. As a result of in vitro study for such a structure, mesenchymal stem cells exhibited a preference for differentiation toward bone lineage [49]. Another often suggested method of apatite deposition on the materials is immersion in simulated body fluid (SBF) [50,51] or similar biomimetic fluids [52]. Coatings fabricated by immersion are precipitated in various topographies, such as flower-, plate-, or needle-like structures [53,54]. However, structures formed by this method often delaminate when submitted to loading [55].

In our previous work, we proved that a promising method of nHA deposition on the surface of polymer fibers is sonocoating. It has already been applied to fabricate nanometric nHA coatings on polymeric scaffolds [56].

The authors of this study designed and fabricated a polymer fibrous scaffold with a controllable degradation rate by coating its surface with nanohydroxyapatite. In the present article, we explore the potential of PDLLA/PLGA electrospun membranes coated with nanoparticles of hydroxyapatite produced using the Microwave Hydrothermal Synthesis (MHS) [57] for Guided Bone Regeneration in periodontal disease. The MHS method permits us to produce unique nHA nanoparticles of precisely controlled grain sizes and narrow particle size dispersion, exhibiting high levels of bioactivity [58]. These features lead to high solubility of the material in TRIS HCl medium in comparison with commercial nanohydroxyapatites. It was shown by Smoleń et al. [59] that MHS method-obtained nHA solubility was 5 times greater than that of commercial nanopowder, but that this property was lost when material was heated. Two coating methods were applied for depositing nHA on fibers: (1) sonocoating of electrospun fibers with nanoparticles, and (2) simultaneous electrospinning of polymer fibers and nHA electrospraying. The coatings produced by the electrospraying method were used as a reference to the sono-deposited coatings. Electrospraying is a method of liquid atomization by electrical forces which does not involve high temperatures. Moreover, during the electrospraying, no additional mechanical energy, other than that from the electric field, is needed for liquid medium atomization. Both methods make it possible to suspend the nHA particles in a liquid medium and deposit them at ambient temperature without influencing their solubility. Sonocoating, especially of bone regrowth scaffolds with nHA, was previously described [56,60]. It was applied to coat various substrates including 3D-printed and electrospun polymers or sintered ceramics [56,60,61,62]. Nevertheless, it is not yet possible to achieve an almost 100% uniform coverage of the electrospun fibers. It is expected that using the nHA particles produced using the MHS method will allow us to achieve a coherent and well-adhering 100–200 nm thick nHA layer, as in the case of 3D-printed polymer scaffolds. The potential of coating in subsequent cycles should be explored as well. Furthermore, the nHA presence effect on membrane biodegradation was never addressed, and will be studied in the present paper. The results of the present study were the basis for a patent application [63], where sonocoating with nHA particles was used for the modification of the surface of non-woven fibrous structures for GTR/GBR procedures in periodontology.

## 2. Materials and Methods 

### 2.1. Materials

Poly(D,L-lactic acid), (PDLLA) with an intrinsic viscosity of 2.0 dL/g and Poly(D,L-lactide-co-glycolide), (PLGA) with a LA/GA molar ratio of 50/50 and an intrinsic viscosity of 1.0 dL/g were purchased from (Polysciences, Inc., Warrington, PA, USA). The 50/50 ratio blend will be further denoted as PDLLA/PLGA. For degradation study, Tris buffer was prepared from Trizma base® pKa 8.06 at 25 °C and Trizma HCl® pKa 8.1 at 25 °C (Sigma-Aldrich, Saint Louis, MO, USA). Two grades of highly-bioactive hydroxyapatite nanopowders for the membrane coating procedures were provided by IHPP PAS, Poland (GoHAP 3 – SSA 186.3 m^2^/g, density 2.93 g/cm^3^, size 15 ± 1 nm and GoHAP 6 - SSA 51.9 m^2^/g, density 3.05 g/cm^3^, size 43 ± 4 nm). The particles have a narrow size distribution and a high degree of crystallinity. The nanoparticles are produced using MHS technology, where the reaction substrates are heated in a pressure vessel using microwaves [56,57,58,64]. They were already successfully applied to coat bone regrowth scaffolds using sonocoating technology [56]. Double distilled water was produced using a water purification system equipped with 0.22 µm filter (HLP 20UV, Hydrolab, Straszyn, Poland). All other chemicals were of analytical grade and used without further purification. For biological studies, the following materials were used: Phosphate-Buffered Saline (PBS) and CellTiter 96^®^ AQ_ueous_ One Solution Cell Proliferation Assay (MTS) (Promega, Madison, WI, USA). Eagle’s Minimum Essential Medium (EMEM) was supplied by the American Type Tissue Culture Collection (ATCC, Rockville, MD, USA) and fetal calf serum was product of Biological Industries (Beit HaEmek, Israel). The cell lines used were human bone osteosarcoma cell line MG-63 and human skin fibroblasts BJ, both purchased from the American Type Culture Collection (ATCC, Rockville, MD, USA) and maintained according to ATCC protocol.

### 2.2. Electrospinning

The membranes were fabricated using the electrospinning technique. PDLLA, PLGA, and polymer blend of PDLLA/PLGA were dissolved in a mixture of solvents, namely chloroform (CHCl_3_) and *N*,*N*-dimethyloformamide (DMF) volume ratio 85/15 to obtain the concentration of 15% w/v. The electrospinning setup consisted of a syringe and needle with an internal diameter of 0.8 mm, a syringe pump (KD Scientific Inc., Holliston, MA, USA), a ground electrode, a stainless-steel rotating drum (200 mm in diameter), and a high voltage power supply. The process was conducted at room temperature, at humidity 40–45%, and at a voltage in the range of 12 kV–14 kV/5 kV at a steady volumetric polymer solution flow rate of 20 μL/min. The randomly-oriented fibers were collected on the rotating drum wrapped with aluminum foil which was kept at a distance of 20 cm from the needle tip and rotated at a speed of 10 Hz (600 rpm). The resulting electrospun membranes were dried overnight under vacuum at room temperature.

### 2.3. Electrospraying

For electrospraying, a mixture of GoHAP 3 and GoHAP 6 nanoparticles in 50/50 ratio was used. Nanohydroxyapatite particles were suspended in methanol to obtain a concentration of 3% w/v, followed by 30 min sonication to disrupt possible agglomerates. The electrospinning setup was combined with electrospraying by vertically attaching a second syringe pump above the collector (see Figure 1). The setup consisted of two syringe pumps, two syringes, and needles with internal diameters of 0.8 mm, a grounded stainless-steel rotating drum (100 mm in diameter), and a high voltage power supply. The process of electrospinning was conducted for 10 min at room temperature, at humidity 40–45%, and with a voltage in the range of 12 kV–14 kV/5 kV. A steady volumetric flow rate of 20 μL/min for electrospinning and 100 μL/min for electrospraying was applied.

### 2.4. Sonocoating 

For the sonocoating procedure, PDLLA/PLGA electrospun membranes were cut into 40 × 80 mm sheets and cleaned in an ethanol:distilled water 1:4 mixture, followed by rinsing with pure distilled water in an ultrasonic bath. Finally, they were dried in a laminar-flow chamber at room temperature for 2 h. The samples were fixed in clasps in a solution of nHA particles in water under an ultrasonic horn according to the patented method of ultrasonic coating (see Figure 2) [60]. Suspensions of 0.5 w/v % concentration in water of nanoparticles GoHAP 3 and GoHAP 6 were prepared separately. The sonocaoting process was performed using a titanium ultrasonic horn Ø 25 mm in diameter at a frequency of 20 kHz. Ultrasonic energy was provided by the VCX750 generator operating at 750 W (Sonics, Newtown, CT, USA). The process was carried out in a sound-absorbing chamber. The temperature of the suspension was kept at 30 °C by means of a cooling system. Two subsequent layers of nanoparticles were deposited in two separate processes. The first layer was composed of GoHAP 6 (43 ± 4 nm size) and was deposited in 5 min, followed by rinsing and drying. Then, the second layer of GoHAP 3 (15 ± 1 nm size) was applied to form a double core-shell coating in a total of 10 min of sonocoating.

### 2.5. Physico-Chemical Characterization of Materials

The morphology of the samples was studied with a Scanning Electron Microscope SEM (S-4800 Hitachi, Tokio, Japan) and Field Emission Scanning Electron Microscope FE-SEM (Ultra Plus GEMINI, Carl Zeiss, Oberkochen, Germany). Prior to the observations, the samples were coated with gold:palladium (80:20) or carbon film with an average thickness of 10 nm to produce conductive surfaces. The observations were run at an accelerating voltage of 2 kV. The structural characteristics of the membranes, such as fiber diameter and pore size, were measured before and after degradation using the ImageJ software (ImageJ v. 1.50i, NIH, Bethesda, MD, USA) collected from 50 random locations. The mean fiber diameter was calculated from SEM micrographs by averaging 25 measurements. Samples after cell culture study were rinsed with PBS to remove non-adherent cells, dehydrated in a series of graded ethanol (50, 60, 70, 80, 90, 95, and 100%), freeze-dried, sputter-coated with gold/palladium (80/20), and observed with SEM.

The chemical structure of the composite membrane was analyzed using Fourier Transform Infrared Spectroscopy (FTIR) (Tensor 27, Bruker, Billerica, MA, USA) equipped with Attenuated Total Reflectance ATR (Platinum ATR-Einheit A 255, Bruker, Germany). The samples were scanned at a temperature of 25 °C and at 15% humidity. A wave number in the range of 4000–650 cm^−1^ and 16 scans per sample were performed for a resolution of 4 cm^−1^. The FTIR analysis of several samples containing nHA, PLGA, PDLLA, PDLLA/PLGA, and PDLLA/PLGA/nHA was carried out.

The content of the HA nanoparticles on the electrospun fibers was measured using thermogravimetric analysis. To record the change in sample mass during heating, samples were measured on a thermogravimetric analyzer (STA 449 F1 Jupiter^®^, Netzsch, Selb, Germany). The samples were heated from 20 to 600 °C at a rate of 10 °C/min under a helium (He) atmosphere at a flowrate of 60 mL min^−1^. Approximately 10 mg of the sample was heated in an alumina crucible.

Because the material composition near the surface was very likely to be different from the bulk, chemical information presented by bulk materials analysis (FTIR) could lead to the misinterpretation of the structure/function relationships. Thus, we performed a surface-sensitive analysis by X-ray photoelectron spectroscopy (XPS) for the elemental surface chemistry of the materials. Materials were studied by X-ray photoelectron spectroscopy (PHI 5000 RSF Versa PRO II, Physical Electronics, Inc., Chanhassen, MN, USA) with a monochromatic AlKα X-ray source. Measurements were performed with an energy resolution of 0.8 eV/step at a pass-energy of 187.85 eV for survey scans and 0.1 eV/step, 23.50 eV pass-energy for high resolution region scans, respectively. As a calibration reference, a carbon peak 1s at 285.0 eV was used to correct for charge effects. Elemental compositions were determined using atom sensitivity factors for carbon (C1s), oxygen (O1s), calcium (Ca2p), and phosphor (P2p). An analysis of the data was performed using the CasaXPS software (CasaXPS v.2.3.15, Casa Software Ltd, Teignmouth, United Kingdom).

The membrane’s wettability was tested by contact angle measurement with HPLC grade water (Chromasol V, Sigma-Aldrich, St. Louis, MO, USA). Tests were conducted on a drop shape analyzer (DSA25, KRÜSS, Hamburg, Germany) device to detect and measure changes in surface wettability from hydrophobic to hydrophilic properties. The water contact angle measurement of samples in air and controlled environment (temperature 25 °C, humidity 50%) were performed with a 5 µL size droplet using a syringe equipped with a needle placed every 10 seconds (10 droplets per sample), and digitally measured with DSA4 software (KRÜSS, Hamburg, Germany) using a geometrical method after the same period of 5 s. The wetting angle was calculated as a mean of 10 measurements ± SD.

### 2.6. In vitro Degradation Study of Membranes 

Electrospun membranes were cut into 1-inch squares with initial weights of 10–12 mg and incubated in 20 mL of 50 mM Tris buffer dissolution medium at 37 °C each. The pH of the buffer solution in ultrapure water with Trizma® base (Sigma-Aldrich, Saint Louis, MO, USA) was adjusted with Trizma® HCl (Sigma-Aldrich, Saint Louis, MO, USA) to pH=7.6 (at 37 °C). To inhibit the growth of bacteria and molds in solution, sodium azide (NaN3) (Mw = 65.02 g/mol, Sigma-Aldrich) was added to Tris buffer. Each type of sample was kept in sealed vessels and incubated in triplicate with the dissolution medium. The pH values of the buffer medium were closely monitored during the experiment. At each degradation time point (0–16 days), specimens were removed from solution and rinsed with deionized water prior to drying in the hood at room temperature for 24 hours and evaluated for weight and molecular mass loss according to ISO norms 10993-13 and 10993-14 [65].

The decrease in polymer molecular weight (Mw) after every degradation time point was studied by gel-permeation chromatography analysis. At each degradation time point (0,2,4,8 weeks), specimens were removed from solution and rinsed with deionized water prior to drying in the hood at room temperature for 24 h. For the measurement, samples were dissolved in a tetrahydrofuran solution (THF) to obtain a 4 mg/mL (w/v) concentration at ambient temperature. THF was also used as a mobile phase eluent for the measurement at flow rate 1 mL/min and injection volume 100 μL at 30 °C. At predetermined degradation time points, samples were withdrawn and passed through a 0.45 μm syringe filter (Nylon, 0.45 μm, Sigma-Aldrich) to remove any non-dissolved polymer and impurities. Viscotek GPCmax (Viscotek, Malvern Panalytical Ltd., United Kingdom) with refractive index (RI) detector and three columns filled with a microporous packing material (PSS SDV 103 Å, PSS SDV 105 Å, and PSS SDV 107 Å) were used. The molecular weights were calculated using polystyrene standard calibration and the OmniSEC 4.5 (Viscotek, Malvern Panalytical Ltd., Malvern, United Kingdom) software. Molecular weight loss was calculated using the following Equation 1, where Mw_1_ and Mw_2_ are the sample molecular weights before and after the degradation time point, respectively. The equation result was expressed as a percentage of the initial Mw.

Mw loss (%) = (Mw_1_ − Mw_2_)/Mw_1_ × 100(1)

To study nanohydroxyapatite particle dissolution in the buffer medium, calcium ion detection by means of Inductively-Coupled Plasma Optical Emission Spectrometry (ICP-OES) was used. Samples of each type were prepared in triplets by placing the electrosprayed and sonocoated membranes of 1-inch squares in 20 mL of TRIS medium in vials. The samples were placed in an environmental shaker at 37 °C and 130 rpm mixing speed. After each time point of degradation (0–16 days) the samples were passed through a 0.45 μm syringe filter to remove possible dust and contaminants (Nylon, 0.45 μm, Sigma-Aldrich). Two percent of HNO_3_ was added to every sample for the dilution procedure. Calcium element values were measured for Ca 396.847 λ and Ca 422.673 λ.

### 2.7. Study of Mechanical Properties

The stiffness and extensibility of the materials were studied by means of uniaxial tensile tests in a wet physiological environment. The thickness of the samples was 100.3 ± 0.9 μm ($T_{\mathrm{mean}}$ ± SD), as measured by analysis of SEM cross-sectional images obtained at n = 45 sample points. The cross-sections of the samples were obtained in the process of immersion in liquid nitrogen for 10 min, followed by scalpel blade cut. The samples were then cut to obtain stripes with length L to width $W_{0}$ dimensions of 80 mm × 10 mm. On each side, 10 mm were used for clamping, leading to a free length of $L_{0}=60\;$mm. Two controlled hydraulic actuators with force sensors (MTS Systems, Eden Prairie, MN, USA) equipped with a CCD camera (Pike F-100B, Allied Vision Technologies GmbH, Stadtroda, Germany) were used for uniaxial tension experiments. Clamping was equipped with SiC sandpaper to prevent undesirable sample slippage. Sample tests were performed at physiological conditions in phosphate buffered saline (PBS) at 37 °C, as the osmolality and ion concentrations of PBS closely match those of the human body. Force data were measured by load cells, and the displacement of the MTS arms during the experiments was recorded simultaneously. The tensile rate for the measurements was 10 mm/min. The nominal stress σ was calculated by dividing the force data F with the reference cross section $A_{0}=W_{0}T_{\mathrm{mean}}$

$$\sigma =\frac{F}{A_{0}}.$$

The distance l of the clamps was related to the reference length $L_{0}$ to calculate the nominal strain:$$\varepsilon =\frac{l-L_{0}}{L_{0}}.$$

### 2.8. Measurements of Cells Metabolic Activity 

The influence of the materials on the proliferation/metabolic activity of MG-63 and BJ cells was measured by MTS assay. For the present test, non-coated and nHA.sonocoated samples were chosen to prove the concept of the novel material for future applications. Briefly, cells were cultured in an EMEM medium supplemented with 10% fetal calf serum (Biological Industries, Israel). Before cell seeding, 96-well microplates (TPP, Trasadingen, Switzerland) containing testing materials were pre-incubated with complete culture media for 24 h. The cells were incubated in a 5% CO_2_ atmosphere at 37 °C. Next, the medium was removed and the plates were rinsed with PBS solution. Both cell lines were seeded in 96-well microplates (TPP Techno Plastic Products AG, Trasadingen, Switzerland) at a density of 1 × 10^4^ cells/well in 100 μL of culture medium. The cells were incubated in plates with membranes for 24 h, 72 h, and 7 days. At least three independent experiments in six replicate wells were conducted per membrane sample type. After the described treatment, 20 μL CellTiter 96 Aqueous One Solution Cell Proliferation Assay (Promega, Madison, WI, USA) was added to each well, and cells were incubated for 3 h. The cells’ metabolic activity was calculated based on absorbance measurements (% cell metabolic activity in the wells with the material compared to the control). The metabolic activity was recorded as relative colorimetric changes measured at 490 nm using a plate reader spectrophotometer Infinite M200 (Tecan GmbH, Grödig, Austria). For statistical analysis, at least three independent experiments were conducted for each toxicity point. The test results for each assay were expressed as a percentage of the untreated control. The difference between samples and the control were evaluated using the GraphPad Prism 5.0 software (Graphpad Software Inc., San Diego, CA, USA). Data were evaluated by Kruskal-Wallis One Way Analysis of Variance on Ranks (ANOVA), followed by the post hoc Dunnett’s method. Differences were considered statistically significant when the p-value was less than 0.05.

## 3. Results

### 3.1. Morphology of the Membranes

An analysis of SEM micrographs revealed that fibers produced from PLGA, PDLLA, and PDLLA/PLGA have a regular texture and smooth surface (Figure 3a–f). Histograms with the diameter frequencies show that the fiber’s thickness was in the range of 2–3 μm.

Electrospraying resulted in the deposition on fibers of GoHAP 6/GoHAP 3 clusters with diameters in the range from 50 to 500 nm (see Figure 4b). On the other hand, the sonocoating of fibers resulted in the homogenous deposition of nanoparticles forming a uniform shell over the polymer fiber core (see Figure 4c–e). A double layer composed of two types of nanoparticles GoHAP 6 and GoHAP 3 was formed homogenously as well, covering 100% of the fiber surfaces (see Figure 4e).

### 3.2. The Chemical Structure Investigated by Means of FTIR

The spectra of the PDLLA, PLGA, and PDLLA/PLGA blend obtained by ATR-FTIR are presented in Figure 5. A sharp peak around 1749 cm^−1^ that appeared in the PLGA, PDLLA, PDLLA/PLGA, and PDLLA/PLGA/nHA composites was assigned to the C=O stretching of polymer. There are also stretching bands due to asymmetric and symmetric C-C(=O)-O vibrations between 1300 and 1150 cm^−1^. Peaks at around 1454 cm^−1^ represent the asymmetric deformation of CH_3_ bonds. Peaks at 1383 cm^−1^ represent δSCH_3_ symmetric deformation, close to 1265 cm^−1^ the stretching modes of ester groups –CO-O-, and around 1085 cm^−1^ the stretching of the C–O–C ether group. The presented spectra characterize important groups of the PLGA and PDLLA molecular structures. The PDLLA/PLGA sample spectrum proves the successful blending procedure by representing bonding features both from the PDLLA and PLGA polymers. The spectra of nanohydroxyapatite powder, as well as the coated polymer composite (PDLLA/PLGA/nHA sonocoated and electrosprayed), exhibit sharp PO4^−3^ bands (characteristic of HAp), which appeared in the regions of 1050 cm^−1^, 603 cm^−1^, and 564 cm^−1^, clearly suggesting the presence of HAp in the membranes.

### 3.3. Elemental Surface Chemistry of Materials (XPS)

The successful grafting of nHA on the surface of the membrane was confirmed by X-ray spectroscopy. The XPS spectra comparison of nHA powders used for coating procedure and natural bone show high levels of elemental similarities (see Figure 6). The presence of nanohydroxyapatite was evident on the coated samples, as transition of the relevant element Ca, P, and oxygen was found (Figure 7). No contaminants were detected. The transition representing calcium and phosphorus or other elements was not detected in the case of unmodified samples, which proves that there were no contaminants on the polymer membrane surface. The binding energies of the Ca2p3/2 and P2p peaks on hydroxyapatite-modified samples were ~345.5 eV and ~132.0 eV, respectively (see Figure 7), which is in agreement with the values for pure nHA in the literature [66]. The results of a table with the elemental concentrations can be found in the Appendix A (Appendix A).

### 3.4. Thermal Properties of the Membranes 

The thermogravimetric analysis of the membranes showed differences in the thermal stability between the tested samples. The thermogravimetry (TG) (see Figure 8a) and differential thermogravimetry (DTG) (see Figure 8b) results indicate that there was an increase in the melting temperature of the material from 360 °C to 395 °C from non-modified PDLLA/PLGA to sonocoated PDLLA/PLGA with the highest content of nHA. Since after polymer evaporation, the residue is hydroxyapatite, the residual weight indicates the nHA amount. The residual weight for the nHA modified samples was 3.6 wt% for electrosprayed samples and 13.6% for the sonocoated, respectively (see Figure 8a).

### 3.5. Wetting Behavior of the Membranes (WCA)

The water contact angle (WCA) was highest for untreated PDLLA (123.1° ± 2.1°), followed by PDLLA/PLGA, and the lowest contact angle (106.9° ± 2.4°) was measured for untreated PLGA (see Figure 9), which is in agreement with literature reports [67,68,69]. Surface modification with electrosprayed nHA caused only a slightly lower contact angle values in comparison with untreated samples. Ultrasonically-coated fibers display highly improved surface wettability; in all cases, the droplets penetrated the membrane matrix in a few seconds. Thus, the ultrasonic coating produced strongly hydrophylic membranes.

### 3.6. Material Degradation and Calcium Ion Concentration 

Degradation studies show a strong influence of coating with nHA on the polymer degradation behavior. The ICP-OES for nHA-coated samples indicate that ultrasonically-deposited particles tend to release calcium ions into the surrounding environment in higher amount and in a more gradual manner than in the case of electrosprayed materials (see Figure 10). pH values during the first 2-weeks of immersion in the medium indicate that the presence of the nHA particles on the surface slowed the rate of degradation of the polymer matrices. SEM images show that electrosprayed particles were deposited on the fibers as agglomerates, while the ultrasonically-deposited ones formed uniform films. For the sonocaoted fibers, segments of the surface where the nanoparticle< coating disappeared degrade more rapidly than segments where the layer is intact (see Figure 10p). Thus, the nHA layer served as a protective layer against degradation (see Figure 10i–p). After 14 days of degradation in the buffer medium at 37 °C of the electrosprayed material, the nHA particles disappeared from the outer regions (see Figure 10a,e). Remnants of nHA were visible only on inner localized fibers. In comparison, for ultrasonically-coated samples, the layer was present on all fibers, almost until the end of the polymer decomposition process (Figure 10i–p), and decompostion started only in the gaps of the nHA layer. 

The initial average Mw values for polymers prior to electrospinning were PLGA Mw = 100–130 kDa, PDLLA Mw = 200–220 kDa, and PDLLA/PLGA Mw = 150–180 kDa, respectively. The molecular weight of the PLGA samples tested by gel-permeation chromatography after 2 weeks’ degradation decreased by over 40%, followed by a gradual loss in sample mass (see Figure 11). The PLGA LA/GA (50/50) material exposed to the buffer medium at 37 °C disintegrated after 8 weeks of degradation, which might be too short a time period to protect the periodontal defect [70,71]. PDLLA/PLGA samples have shown moderate molecular weight loss during the degradation process (around 50% loss for an 8-week degradation period). As degradation proceeded, the reduction in molecular weight led to a decrease in the physical properties by the formation of water-soluble polymer fragments. These water-soluble chains diffuse away from the polymer, and are ultimately hydrolyzed to glycolic and lactic acids, which can be processed through normal metabolic pathways.

The nHA layers produced by means of sonocoating protect the polymer fibers from degradation. The only areas that started to degrade were those where the layer detached or dissolved, leaving the polymer surface exposed to the test fluids. From Figure 10 and Figure 11, it is clearly seen that the electrosprayed layer nHA has little effect on the polymer degradation process, while the sonocoated layer visibly prevents degradation.

### 3.7. Mechanical Properties of the Membranes

The mechanical behavior of membranes was measured in wet (see Figure 12) physiological conditions (PBS medium) to investigate their future functionality in the human body. Polymers used for the present study have shown a linear stress–strain response before yielding [72,73]. Neither type of surface modification caused major material structure weakening. Sonocoated samples in wet conditions displayed a stress response in the same order of magnitude as commercially-available GTR membranes [74,75] and, more importantly, in the same order of magnitude as living periodontal tissues [76,77]. Figure 12b–d depict the sample morphology after the test. Interestingly, the electrosprayed nHA particles detached from the fibers during tensile test (see Figure 12c). In contrast, the layer of sonocated particles remained attached to the fibers during deformation (Figure 12d), and the nHA coating was present until the end of the tensile test in PBS. Figure 12e–f depicting the optical images of samples before and during the test show visual differences in the samples’ wetting behavior.

### 3.8. In Vitro Cytotoxicity to MG-63 and BJ Cells

The cytotoxic activity of PDLLA/PLGA and PDLLA/PLGA/nHA sonocoated membranes were investigated on the human osteosarcoma cells (MG-63) and skin fibroblastic cells (BJ) using the MTS assay. Graphs of metabolic activity % in Figure 13a,d indicate that the membranes did not have any toxic effect on the osteoblast-like and fibroblastic cells, but increased the viability of cells. As shown in Figure 13d, metabolic activity of MG-63 cells on the surface of PDLLA/PLGA, and PDLLA/PLGA/nHA sonocoated membranes were similar after the first 24 h. However, after 7 days of cell culture, the metabolic activity varied between samples, and significantly increased mitochondrial activity was observed for the PDLLA/PLGA/nHA sonocoated membranes. In contrast, seeding BJ cells (Figure 13a) on PDLLA/PLGA/nHA sonocoated membranes causes an increase in cell proliferation (350% after 7 days), but it was not as high as with MG-63 cells (almost 600% of metabolic activity).

As seen in Figure 13f, the MG-63 cells were strongly anchored on the nHA sonocoated samples, with preferential attachment to the fibers of high surface.

## 4. Discussion 

The present study addresses the issue of optimizing the membrane design for GBR/GTR treatment of periodontal defects. A novel method for electrospun membrane modification with nHA by means of two-step sonocoating was compared with the one-step electrospraying method. The results have shown that both methods of are non-destructive and effective for the surface modification of the electrospun constructs. Electrospraying is a one-step process, whereas ultrasonic coating requires the deposition of nanoparticles after electrospinning. However, it was reported that the electrospraying of non-woven structures is very difficult to scale-up when industrial manufacturing is considered [78,79]. In contrast, sonocoating is already used for the surface modification of textiles with antibacterial nanoparticles on a large industrial scale [80], and scaling up this technology for the present membranes is possible as well. 

The sonocoating method allowed us to obtain membranes with the unique features required for GTR/GBR applications. Several materials commercially used for biodegradable membranes generally lose mechanical strength and stop acting as a physical barrier separating tissues soon after the implantation procedure [5,6,8,10,11,12,13,14]. When degradation progresses, the material is still physically present between tissues, but its tissue separation function is lost quickly. The rate at which a GTR membrane degrades plays an essential role in tissue regeneration, as the rate of fiber degradation should match the rate of new tissue formation. Therefore, it is crucial to be able to control the degradation process of fibrous membranes. The present paper shows that surface nHA coatings may help to tailor scaffolds for periodontal applications. Therefore, the sonocoated membranes proposed in this study are promising for the treatment of periodontal tissue regeneration. The presented membranes did not have any toxic effect on the osteoblast-like or fibroblastic cells, and increased the viability of cells for up to 7 days of culture. The MG-63 (osteosarcoma) cells on the surface of PDLLA/PLGA/nHA sonocoated membranes displayed the highest metabolic activity among the tested materials. Seeding BJ (fibroblasts) cells on PDLLA/PLGA/nHA sonocoated membranes caused an increase in cell proliferation, but this was not as large as with MG-63 cells. This was expected, since the coating should foster the growth of bone tissue. Thus, the fiber coating made from highly-biocompatible nanohydroxyapatite produced using the microwave MHS method used in the present study provided an environment which is potentially friendly for bone cells. 

The application of a bioactive nHA coating dramatically increased the surface wettability of the materials. Moreover, the sonocoating method was proved to be suitable for obtaining homogenous coatings composed of subsequent nanoparticles layers, even on thermally-sensitive materials such as electrospun biodegradable polymers. This is due to mild process conditions: coating in water at a temperature of 30 °C and for a period of 10 min. 

Apparently, the increase in wettability of the sonocoated samples suggests that water molecules have easy access to the material surface, and could accelerate the degradation of polymers. However, the nHA layer caused strong hydrophilicity of the membranes while acting as a protective layer. 

It seems that the nHA coating layer degradation takes place according to two different mechanisms for the electrosprayed and sonocoated layer, respectively (see Figure 14). The electrosprayed particles are only weakly bonded to the polymer surface, and easily detach, leading to a release of the nanoparticles (see Figure 14). Subsequently, degradation of the whole polymer surface takes place. The sonocoated layer is coherent and strongly bonded to the polymer surface. It serves as a protection layer against polymer degradation. In places where gaps in the layer eventually appear (due to nanoparticles dissolution or detachment), polymer degradation occurs, as seen by the formation of cavities on its surface.

An important factor reported in the literature of periodontal disease treatment is the presence of acidic degradation by-products of PLGA and PDLLA copolymers, which may accumulate around a surgical site, cause an inflammatory reaction, and negatively affect the bone tissue regeneration process [81]. In degradation tests, the nHA particle coating reduced the acidity of the degradation medium. The calcium ions eluded from the hydroxyapatite layer on polymer fibers could contribute to the neutralization of the acidic polymer degradation byproducts. In case of electrosprayed samples, a drop in pH during degradation was minimized over a relatively short period of time (see Figure 10). On the other hand, for the sonocoated samples, acidity was reduced more strongly than for the electrosprayed samples and over a relatively long period of time. This is probably due to a higher level of calcium ion release from sonocoated samples than for the electrosprayed ones, which is in agreement with the higher number of nanoparticles deposited on the polymer surface. 

The proposed mechanism of deterioration of the nHA coating is shown in Figure 14. It is suspected that the nHA layers produced by means of sonocoating attached firmly to the fibers, forming a uniform layer. This strong attachment may be attributed to the lateral growth of the nanoparticle layer during the process of sonocoating previously reported by Woźniak et al. [62]. Areas where the layer detached or dissolved and the polymer was exposed to the test fluids started to degrade the prior nHA covered surfaces. 

The possible reason why nHA acts as a protective layer is that nanohydroxyapatite particles sonocoated on fibers may hinder water diffusion towards the polymer fibers and slow down the hydrolytic cleavage. Moreover, nHA may prevent the diffusion of intermediate degradation products out of the polymer, slowing down the pH drop. This is ascribed to the reaction of calcium ions and the carboxyl end-groups generated during the degradation of the polymer to form calcium carboxylate chain ends. As a result, the acidic autocatalytic degradation of the polymer is slowed down, so that possible inflammation reactions could be avoided or be less severely present [82]. 

For a wet uniaxial tensile test in the PBS medium, all membranes presented satisfactory mechanical response, i.e., comparable with the most common commercial membranes [74,75] and stresses in the same order of magnitude as the native tissue [76,77]. Nevertheless, they exhibited lower values than the stress–strain responses of commercial materials measured in dry conditions [83]. The nHA layer produced using sonocoating did not delaminate during deformation. The nanoparticles were still attached to the fibers after the test, but did not form a continuous layer. 

This study provided the first evidence that nanohydroxyapatite deposited by means of sonocoating could potentially improve bone cells adhesion, structural stability, harmful pH variations, and wetting properties, and could extend the degradation time of a membrane for possible application in the treatment of periodontal disease. 

## 5. Conclusions

Biodegradable membranes for potential use in periodontal tissue defect regeneration with precisely controlled compositions and programed degradation times were produced by means of the electrospinning method followed by sonocoating with nanohydroxyapatite particles. 

Sonocoating permitted the creation of a uniform, fully covering layer of nanohydroxyapatite (nHA) on the surface of the electrospun polymer fibers. It was proved that the formation of a double layer, i.e., the inner one made from 43 ± 4 nm particles and the outer from 15 ± 1 nm particles, was successful, and led a change in the properties of the fibrous material. Coating by the ultrasonic method considerably extended the degradation time of the membranes and decreased the acidity caused by the hydrolytic degradation around the material. The nHA layer transformed the hydrophobic PDLLA/PLGA membrane into a hydrophilic one. The presented membranes did not have any toxic effect on the osteoblast-like or fibroblasts cells, and increased the viability of the cells during a 7-day culture. The cells’ metabolic activity was most pronounced for the PDLLA/PLGA/ nHA sonocoated samples and after 7 days of MG-63 osteosarcoma cells culture.

The sonocoated membranes in a wet state displayed mechanical properties that were comparable to those of other, commercially-available GTR membranes; more importantly they matched the mechanical properties of living periodontal tissue. Thus, such barrier membranes are promising biocompatible materials for the regeneration of cavities in periodontal tissue. 

## 6. Patents

Institute of High Pressure Physics PAS is an official holder of “Method of manufacturing bone implant and bone implant” - Polish granted patent PL226891 (B1), European Patent Granted EP3291850 (B1), USA Patent Application US 2018/0311407 (A1), International Patent Application WIPO/PCT WO2016178174 (A1), Polish Patent Application “Biological Barrier Membrane” PL427554 (A1) and International Patent Application PCT/PL2019/050057.

## Figures and Tables

**Figure 1 nanomaterials-09-01625-f001:**
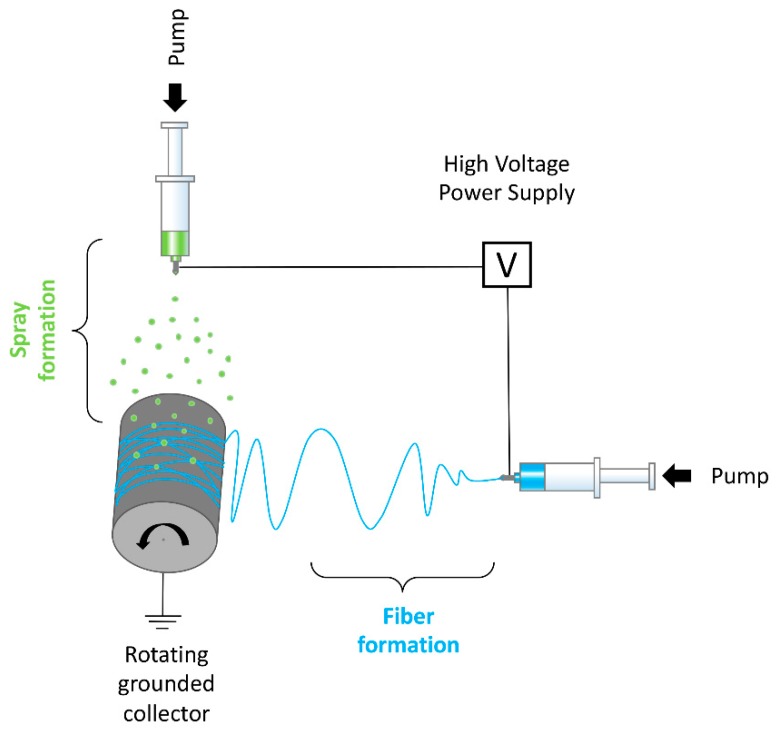
Electrospinning setup combined with electrospraying.

**Figure 2 nanomaterials-09-01625-f002:**
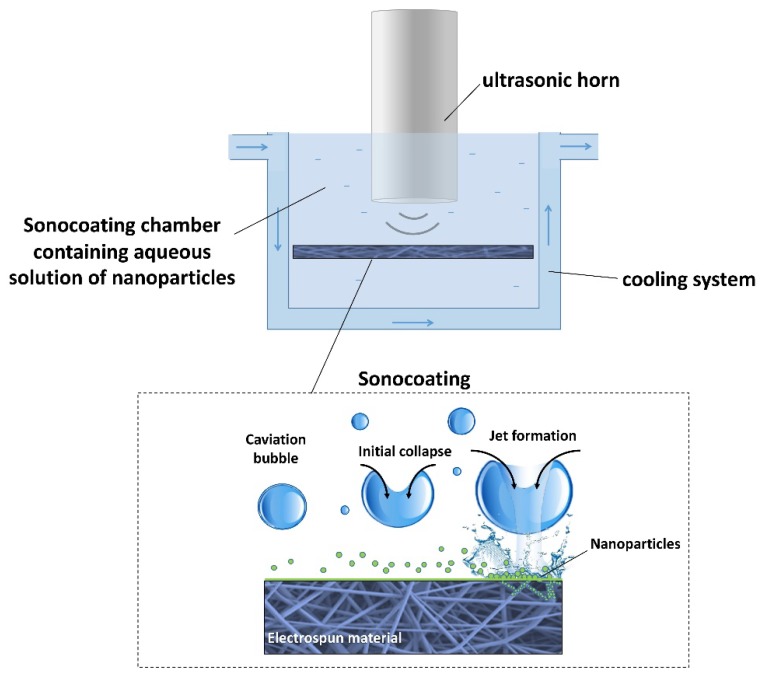
Schematic view of ultrasonic coating setup and ultrasonic cavitation phenomenon occurring near the membrane surface during the process.

**Figure 3 nanomaterials-09-01625-f003:**
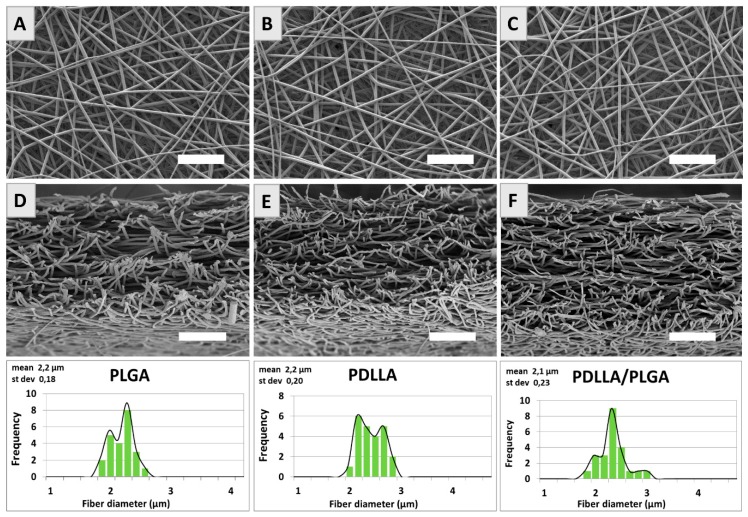
SEM images of PLGA, PDLLA and PDLLA/PLGA samples top view (**A**–**C**) and fractured in liquid nitrogen cross-sections (**D**–**F**). For each material, fiber diameter histograms are given. Scale bar is 50 μm.

**Figure 4 nanomaterials-09-01625-f004:**
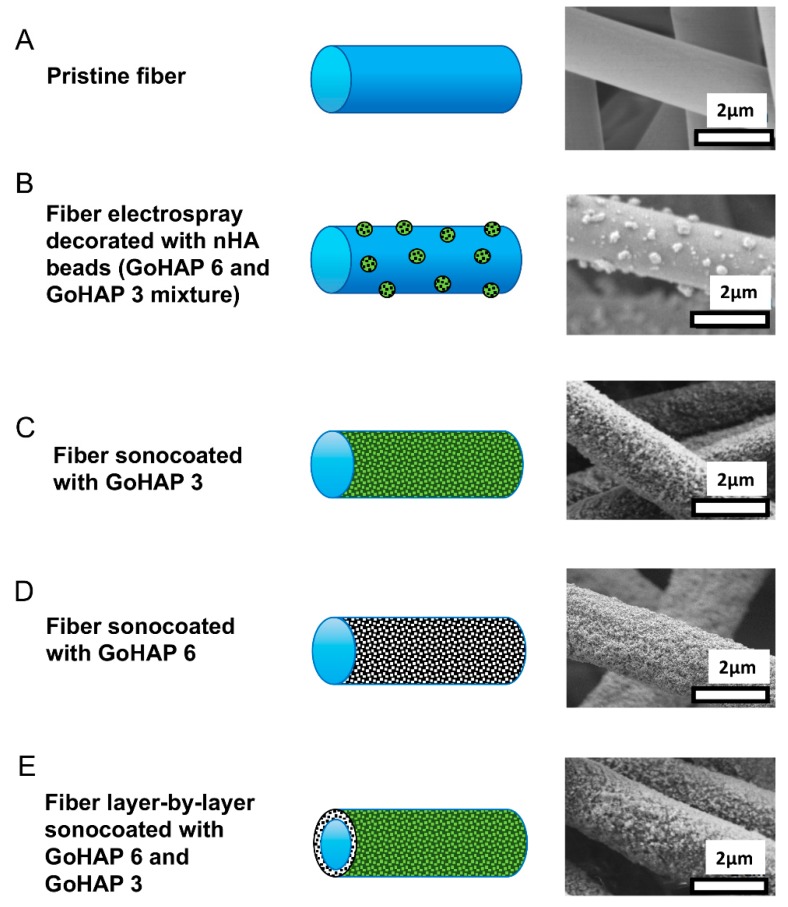
SEM images and morphology scheme of PDLLA/PLGA membrane fibers: (**A**) Pristine PDLLA/PLGA fiber; (**B**) fiber electrospray covered with nHA beads; (**C**) fiber sonocoated with GoHAP 3; (**D**) fiber sonocoated with GoHAP 6; (**E**) fiber layer-by-layer sonocoated with GoHAP 6 and GoHAP 3, respectively.

**Figure 5 nanomaterials-09-01625-f005:**
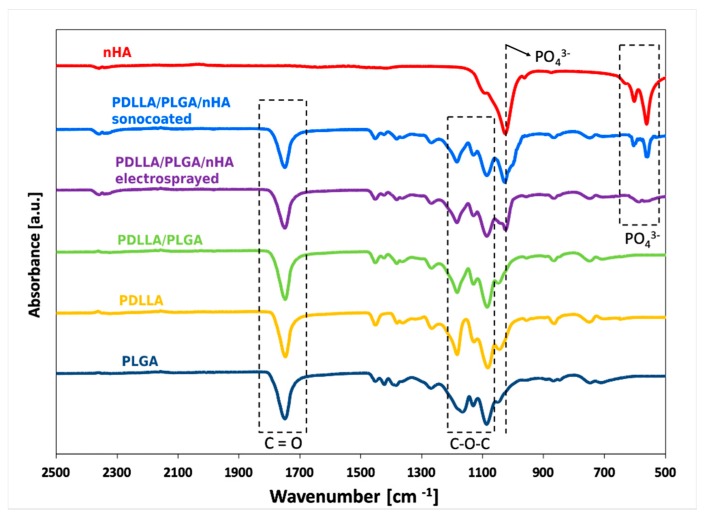
FTIR spectra of a: nHA, PDLLA, PDLLA/PLGA, and PDLLA/PLGA coated with nHA.

**Figure 6 nanomaterials-09-01625-f006:**
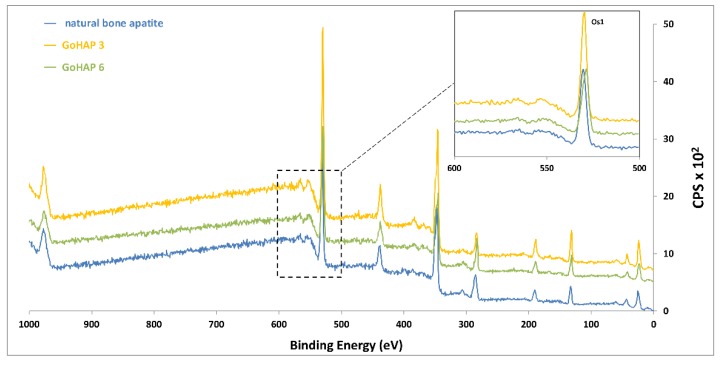
XPS spectra of GoHAP 3 and 6 powders compared with natural bone apatite.

**Figure 7 nanomaterials-09-01625-f007:**
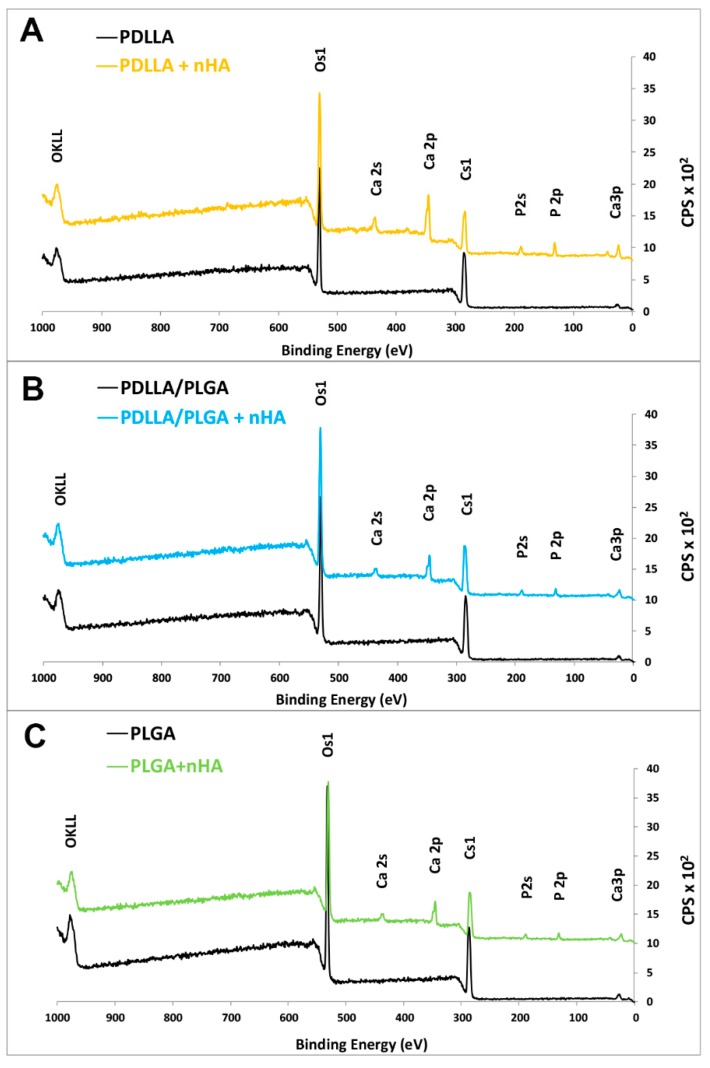
XPS spectra of unmodified and nHA double-layer sonocoated materials: (**A**) PDLLA; (**B**) PDLLA/PLGA; and (**C**) PLGA.

**Figure 8 nanomaterials-09-01625-f008:**
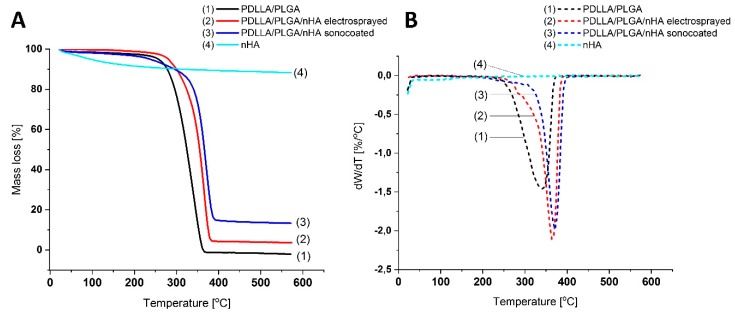
TG (**A**) and DTG (**B**) curves at heating rate b = 10 °C min^−1^ of a PDLLA/PLGA (1), PDLLA/PLGA/nHA electrosprayed (2), PDLLA/PLGA/nHA sonocoated, and (3) and nHA (4).

**Figure 9 nanomaterials-09-01625-f009:**
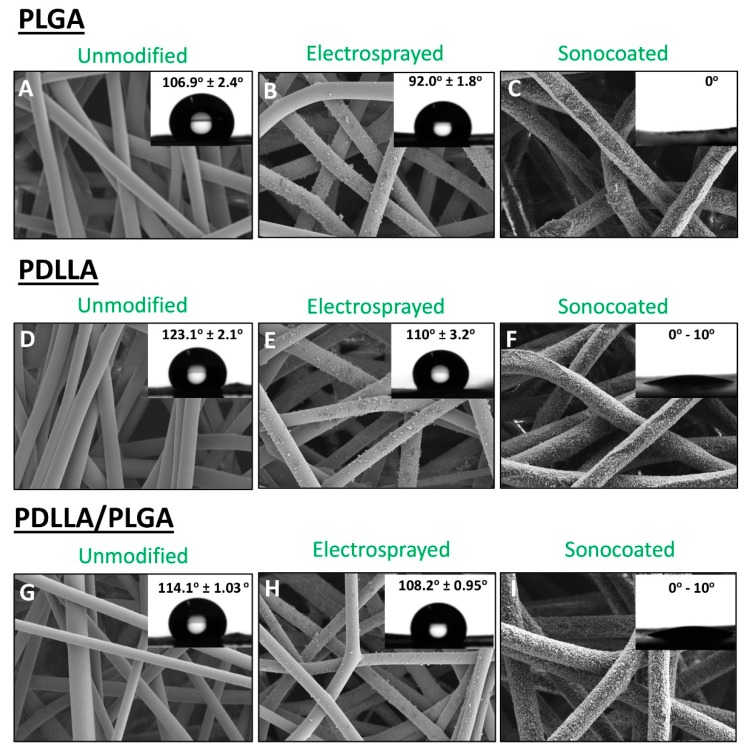
SEM images and water contact angle measurement values of materials and composites: (**A**–**C**) PLGA; (**D**–**F**) PDLLA; (**G**–**I**) PDLLA/PLGA.

**Figure 10 nanomaterials-09-01625-f010:**
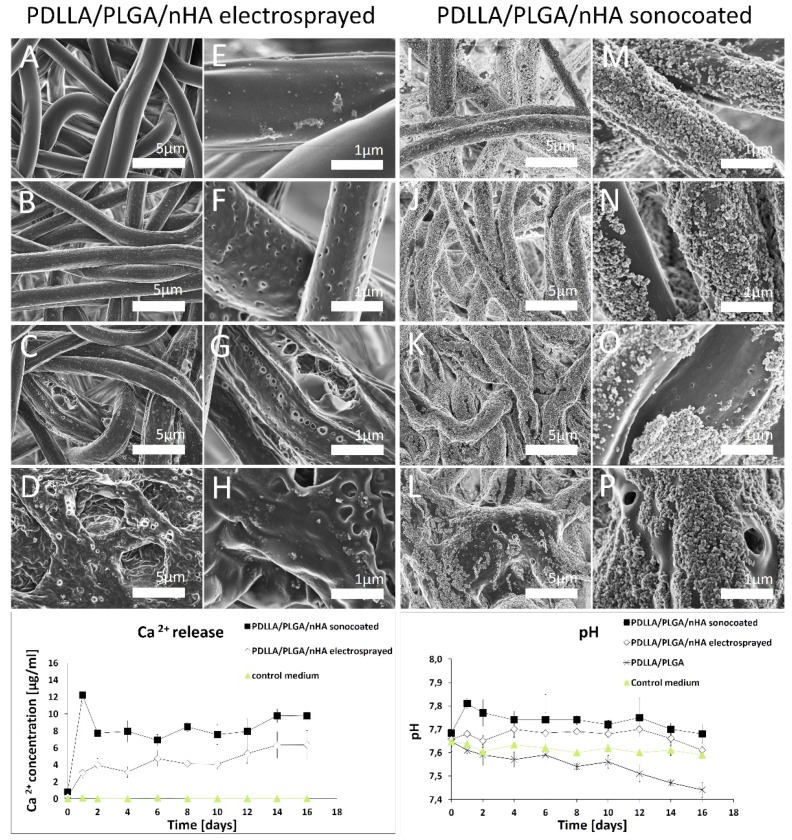
Graphs of ICP-OES calcium release and pH changes profiles, and SEM images of PDLLA/PLGA/nHA Electrosprayed fibers, (**A**–**H**) PDLLA/PLGA/nHA Sonocoated fibers (**I**–**P**), after 2,4,8,10 weeks of degradation (from the top down, respectively).

**Figure 11 nanomaterials-09-01625-f011:**
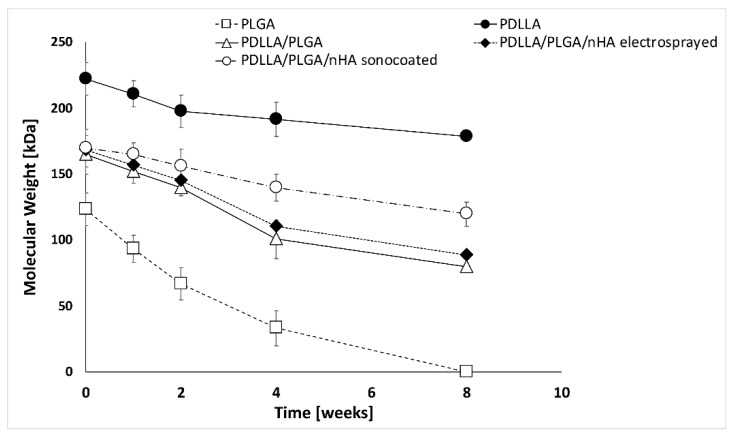
Graph of molecular weight (Mw) changes in the function of degradation time.

**Figure 12 nanomaterials-09-01625-f012:**
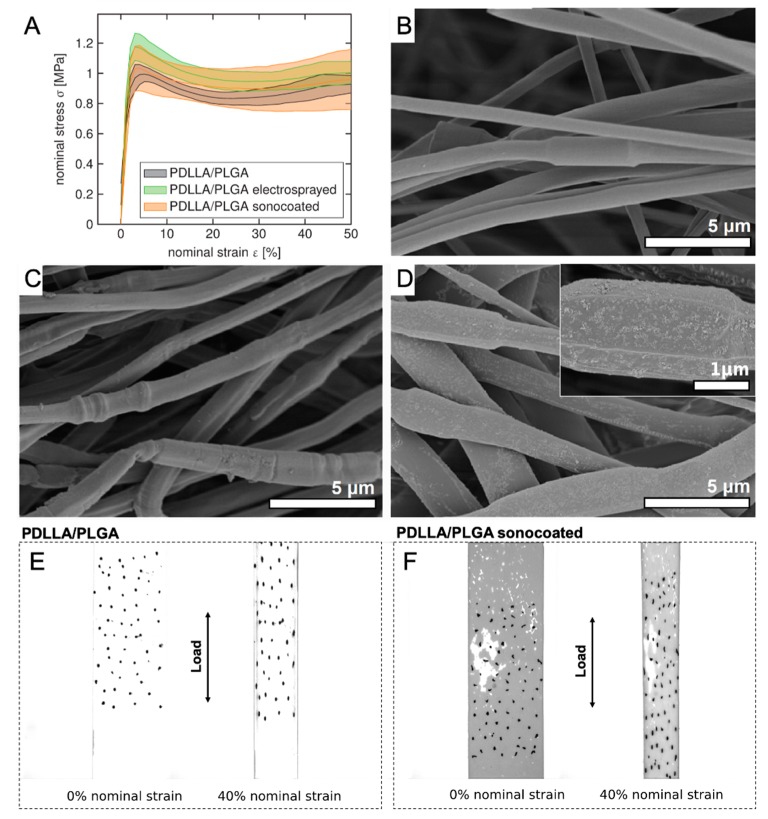
(**A**) Nominal strain-stress curve for coated and non-coated samples. All graphs show mean and standard deviation (shaded areas) from *n* = 3. SEM images of samples after tensile tests: (**B**) PDLLA/PLGA; (**C**) PDLLA/PLGA/nHA electrosprayed, and (**D**) PDLLA/PLGA/nHA sonocoated. Optical CCD camera images of (**E**) PDLLA/PLGA and (**F**) PDLLA/PLGA/nHA sonocoated during tensile properties testing in PBS.

**Figure 13 nanomaterials-09-01625-f013:**
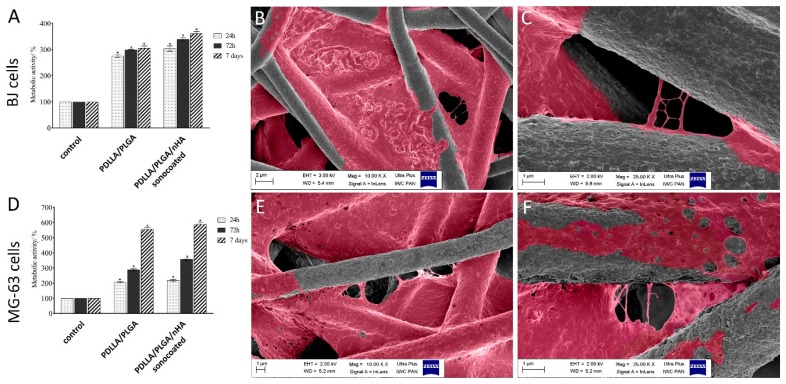
Metabolic activity of BJ (**A**) and MG-63 (**D**) cells seeded on PDLLA/PLGA and PDLLA/PLGA/nHA sonocoated materials after 24 h, 72 h, and 7 days. SEM colorized images of MG-63 cells attachment after 24 h on (**B**,**C**) PDLLA/PLGA and (**E**,**F**) PDLLA/PLGA/nHA sonocoated materials. (**B**,**E**): magnification 10kx; (**C**,**F**): magnification 25kx. Metabolic activity expressed as a percentage of the control, mean ± SD from three independent experiments. (*denotes statistically significant difference from unexposed control, *p* < 0.05).

**Figure 14 nanomaterials-09-01625-f014:**
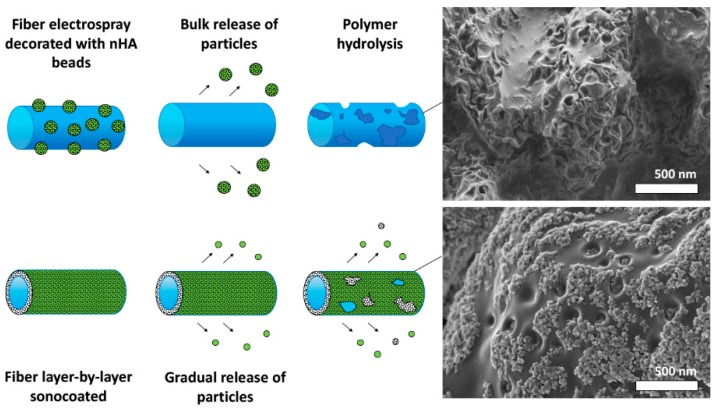
Scheme of particles release mechanism and SEM images of Electrosprayed (**top**) and Sonocoated (**down**) PDLLA/PLGA fibers after 10 weeks of degradation.

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
