# Peer review of "Polymer Membranes Sonocoated and Electrosprayed with Nano-Hydroxyapatite for Periodontal Tissues Regeneration"

_nanomaterials, 2019, doi:10.3390/nano9111625_

Round 1
Reviewer 1 Report
All comments have been taken into account and well answered.
Thus, the article could be published in this form.
Author Response
Dear Reviewer,
We hereby submit the revised version of the manuscript initially entitled “Polymer Membranes Sonocoated and Electrosprayed with Nano-hydroxyapatite for Periodontal Tissues Regeneration” for consideration to be published in Nanomaterials Journal (Special Issue: Nanomaterials and Nanotechnology in Dentistry). We appreciate all previous comments and suggestions. We have addressed the issues of minor English language editing raised by both Reviewers. Thanks to valuable comments the quality of the manuscript text was highly increased.
Sincerely Yours,
Julia Higuchi
Corresponding Author

Reviewer 2 Report
The authors adequately addressed the previous concerns. However, several concerns were identified in the revised text.
Major Concerns:
Line 141: The “high solubility” of these nHA particles should be stated in the context of the solvent. What is the solvent specifically - is it methanol?
Lines 494-495: Provide a citation to the literature supporting the claim that the polymers used in this study have shown a linear stress-strain relationship.
Line 499: How was the “final point before breaking” determined? It is not entirely clear how this was determined since the stress-strain data presented suggests plastic deformation rather than failure.
Line 502: The images used in Figure 12 B-D should focus on regions where the deformation occurred (the close-up view showed is focused on a part of the fiber that was not deformed). Furthermore, it is not clear what the authors are trying to convey with Figure 12 E and F since there is no description of what the markings on the optical CCD images are and these panels are not discussed in the main body of the manuscript.
Line 591: Should be changed to "over a relatively longer period of time." Additionally, is this difference statistically significant?
In addition, the following issues were also noted during the new round of reviewing:
Lines 517-520 and 556-557: MG-63 are a cancer cell line that proliferate much faster than human osteoblasts, so it makes sense that they would proliferate faster than a fibroblast line. Thus, it is trivial to state that MG-63 cells proliferate faster than the fibroblast line.
Lines 521 and 631: Figure 13 only shows data for cells on PDLLA/PGA and PDLA/PGA/nHA sonocoated samples - there is no data for electrosprayed samples. This is a bit misleading.
Lines 623-624: The diameter of the outer layer cannot be smaller than the diameter of the inner layer, therefore, one or both of those parameters is supposed to be thickness of the layer not diameter. Both of these are referred to as thickness in lines 231 and 232.
Minor Comments:
Mistakes in grammar and word choices are numerous - sometimes making the information unclear. Below are a list of specific examples:
Lines 30 and 52: The definition of nHA is not consistent. Is it nanohydroxyapatite or hydroxyapatite nanoparticles? These seem to be used interchangeably.
Lines 94-95: This sentence is important, and the mechanical properties of these membranes is important, but this sentence does not really say anything about them being suitable for this application or what “good mechanical properties” actually are.
Lines 117-118, 355, 631: Possessives are traditionally not used in scientific writing.
Line 122: An abbreviation is given for human fetal osteoblastic (hFOB) cells, but the cells are never mentioned again in the manuscript.
Lines 123-124: This sentence seems kind of vague as it is not clear exactly what is meant by the group's “successful fabrication” of a tissue engineering scaffold. Or is it just important that they fabricated a scaffold with a high amount of surface roughness in the nanometer range using an electro spraying technique? Ramier et al. showed that incorporation of nHA within/on their PHB fibers caused changes in their mechanical properties and cell response.
Line 124: Poly(3-hydroxybutyrate) is referred to only by its abbreviation PHB, but it is not a common enough polymer to give the abbreviation without further explanation.
Lines 126-127: This sentence does not make sense. It is unclear what the point the authors are trying to convey.
Line 127: “Apatites” should not be plural here.
Line 136: “Hydroxyapatite nanoparticles” should just be referred to as nHA. Depending on the definition of nHA.
Lines 137-138: “Nanoparticles of hydroxyapatite” should just be referred to as nHA. Depending on the definition of nHA.
Line 142: “MHM” is used instead of “MHS”.
Lines 147 and 149: The word “atomisation” is used, but the correct word is “atomization”.
Line 151: The word properties should be removed form the end of the sentence.
Line 151-153: Authors should not be referenced by number in this manner, keep it consistent with other portions of the manuscript like this within the text.
Line 156: “MHH” is used instead of “MHS”
Lines 158-159: This sentence is a big vague and worded in a confusing manner.
Lines 173-174: The sentence should read “…substrates are heated in a pressure vessel…” instead of “hearted”.
Line 224: 0,5 w/v% should be written as 0.5 w/v% to maintain consistency throughout the manuscript.
Line 304: The abbreviation “Mw” was defined on line 294.
Line 312: The full name for the abbreviation ICP-OES is never given.
Line 337: “Nominal stretch” should be called strain. Which is how it is defined in Figure 12 A.
Line 334: The word “deviding” should be spelled “dividing”.
Line 375: SEM was previously defined so it should be used here, also the beginning of this sentence could be rewritten as "Analysis of SEM micrographs revealed..."
Line 432: No parentheses around TG and DTG abbreviation.
Line 442: 123,1° ± 2,1° should be 123.1° ± 2.4° to maintain consistency within the manuscript.
Line 443: 106,9° ± 2,4° should be 106.9° ± 2.4° to maintain consistency within the manuscript.
Line 461: Should change "fibres" to "fibers" for consistency.
Line 475: The reference to Figure 11 is only given with respect to PLGA, but the figure has much more information than just those samples.
Line 495: The word “types” should not be plural here.
Line 496: Instead of referring samples as “ultrasonically coated” they should be referred to as “sonocoated” for consistency.
Lines 525-526: Magnification should not be abbreviated in line 526, you must maintain consistency.
Lines 535-537: This is not a complete sentence.
Lines 558-560: Not all lines that have been changed since the last review have been highlighted, this is the one of those.
Line 572: This should be “weakly” instead of “weekly”.
Line 589: Remain consistent with the nomenclature that has been used in this paper previously, namely that these are called nHA particles.
Lines 592-594: This sentence does not really make sense.
Line 595: “Form” should be changed to “forming”.
Line 602: “Sono-deposited” should be changed to sonocoated since that is what they have been called up until this point.
Lines 614-617: The word “the” in “…sonocoating could potentially improve the bone cells adhesion” does not need to be there. This sentence needs some work.
Line 626: The short sentence here does not really contribute to the conclusion of the manuscript.
Author Response
Dear Reviewer,
We hereby submit the revised version of the manuscript initially entitled “Polymer Membranes Sonocoated and Electrosprayed with Nano-hydroxyapatite for Periodontal Tissues Regeneration” for consideration to be published in Nanomaterials Journal (Special Issue: Nanomaterials and Nanotechnology in Dentistry). We appreciate all comments and suggestions. We have addressed them below raised point by point. The responses are provided and written in blue and changes were applied according to the suggestions in the final version of the text.
Major Concerns:
Line 141: The “high solubility” of these nHA particles should be stated in the context of the solvent. What is the solvent specifically - is it methanol?
Yes, indeed we did specified what was shown in given reference study [Smoleń et al. ref. 58]. Rewritten to: These features cause high solubility of the material in TRIS HCl medium in comparison with commercial nanohydroxyapatites.
Lines 494-495: Provide a citation to the literature supporting the claim that the polymers used in this study have shown a linear stress-strain relationship.
Supporting our claim we have added literature reports confirming the initial linear response detectable and measurable up to very low strain (0.1–0.2%):
Kolluru, P.V.; Lipner, J.; Liu, W.; Xia, Y.; Thomopoulos, S.; Genin, G.M.; Chasiotis, I.; Strong and Tough Mineralized PLGA Nanofibers for Tendon-to bone Scaffolds. Acta Biomaterialia, 2013, 9(12), Pages: 9442-9450.
Sweeney, J.; Spencer, P.; Nair, K.; Coates, P.; Modelling the Mechanical and Strain Recovery Behaviour of Partially Crystalline PLA. Polymers (Basel), 2019, 11(8), Pages: 1342
Line 499: How was the “final point before breaking” determined? It is not entirely clear how this was determined since the stress-strain data presented suggests plastic deformation rather than failure.
We have re-written the following sentence to: Figures 12B-D depicts the samples morphology after the test.
Line 502: The images used in Figure 12 B-D should focus on regions where the deformation occurred (the close-up view showed is focused on a part of the fiber that was not deformed). Furthermore, it is not clear what the authors are trying to convey with Figure 12 E and F since there is no description of what the markings on the optical CCD images are and these panels are not discussed in the main body of the manuscript.
In our opinion the Figures 12E and 12F are accurately depicting the overview of samples wetting behavior also before and during the wet tensile test. Thus, we have added the following description to the body of the text:
Figures 12E-F depicting the optical images of samples before and during the test show visual differences in samples wetting behavior.
Line 591: Should be changed to "over a relatively longer period of time." Additionally, is this difference statistically significant?
The sentence was re-written according to the suggestion:
The effect of the drop in pH during the degradation was minimized over a relatively longer period of time in the case of coating in the elecrospraying process (Figure 10).
In addition, the following issues were also noted during the new round of reviewing:
Lines 517-520 and 556-557: MG-63 are a cancer cell line that proliferate much faster than human osteoblasts, so it makes sense that they would proliferate faster than a fibroblast line. Thus, it is trivial to state that MG-63 cells proliferate faster than the fibroblast line.
MG-63 is indeed an osteosarcoma cancer cell line. Nevertheless, in cytotoxicity studies of bone regeneration materials they are widely used and often compared with fibroblastic cell lines. There are many literature reports where MG-63 cells are used for cytotoxicity studies with fibroblast cells (e.g. murine NIH-3T3 cells) [a,b,c]. The referred studies present scaffolds for supporting growth of various cell lines, such as: murine aneuploid fibro sarcoma (L929), human osteosarcoma cells (MG-63), and mouse embryo fibroblasts (NIH3T3), exhibiting prospective application both in skin and bone tissue engineering. Fibroblasts in contact with electrospun materials containing calcium phosphates exhibit comparable results to MG-63 cells in short term studies (0-72h) both in cytotoxicity, proliferation and spreading. Nevertheless, the cytocompatibility studies using MG-63 cells in comparison with NIH-3T3 fibroblasts can show an increase in the cell viability with respect to longer culture time. Thus, in our opinion the preliminary in vitro study (0-72h) shown in our manuscript is supporting the claim of favorable MG-63 osteoblastic response towards nHA coated samples. Of course we have in mind that the most relevant primary cells in the bone tissue microenvironment are bone marrow stromal cells. Therefore, it is necessary to further study the in vitro cell biocompatibility of presented materials using the most relevant primary cells. However, our work has initially demonstrated that studied composites have good in vitro cell biocompatibility.
[a] Liu, X.; Wei, D.; Zhong, J.; Ma, M.; Zhou, J.; Peng, X.; Ye, Y.; Sun, G.; He, D.; Electrospun Nanofibrous P(DLLA–CL) Balloons as Calcium Phosphate Cement Filled Containers for Bone Repair: in Vitro and in Vivo Studies. ACS Applied Materials & Interfaces, 2015, 7(33), Pages: 18540-18552.
[b] Chaurey, V.; Block F.; Su, Y.H.; Chiang, P.C.; Botchwey, E.; Chou, C.F.; Swami N.S.; Nanofiber size-dependent sensitivity of fibroblast directionality to the methodology for scaffold alignment. Acta Biomaterialia , 2012, 8(11), Pages: 3982-90.
[c] Shalumon, K.T.; Anulekha, K. H.; Chennazhi, K. P.; Tamura, H.; Nair, S. V.; Jayakumar, R.; Fabrication of chitosan/poly(caprolactone) nanofibrous scaffold for bone and skin tissue engineering. International Journal of Biological Macromolecules, 2011, 48(4), Pages: 571–576, 2011.
Lines 521 and 631: Figure 13 only shows data for cells on PDLLA/PGA and PDLA/PGA/nHA sonocoated samples - there is no data for electrosprayed samples. This is a bit misleading.
Yes, indeed we did eliminated the electrosprayed samples from the cytotoxicity study and the justification to use electrospraying as a comparative methods was already well explained in the introduction section. We have added also additional sentence to the Methods section: For the present test non-coated and sonocoated samples were chosen to prove the concept of novel ultrasonically modified material qualification for future application.
Lines 623-624: The diameter of the outer layer cannot be smaller than the diameter of the inner layer, therefore, one or both of those parameters is supposed to be thickness of the layer not diameter. Both of these are referred to as thickness in lines 231 and 232.
The values are correct. They are not describing the layer thickness but the size of particles used. Thus, we have re-written the following sentence for better understanding:
Lines 230-232: The first layer was composed of GoHAP 6 (43 ± 4 nm particles size) and was deposited in 5 min, followed by rinsing and drying. Then the second layer of GoHAP 3 (15 ± 1 nm particles size) was applied to form a double core-shell coating in a total of 10 min of sonocoating.
Minor Comments:
Mistakes in grammar and word choices are numerous - sometimes making the information unclear. Below are a list of specific examples:
Lines 30 and 52: The definition of nHA is not consistent. Is it nanohydroxyapatite or hydroxyapatite nanoparticles? These seem to be used interchangeably.
We have chosen nanohydroxyapatite to be used along the whole article text for consistency.
Lines 94-95: This sentence is important, and the mechanical properties of these membranes is important, but this sentence does not really say anything about them being suitable for this application or what “good mechanical properties” actually are.
We did specified, lines 94-96: Electrospun membranes for load-bearing applications in implantology need to exhibit sufficient mechanical properties to allow placement in defect, avoiding membrane collapse and providing barrier function.
Lines 117-118, 355, 631: Possessives are traditionally not used in scientific writing.
Re-written
Line 122: An abbreviation is given for human fetal osteoblastic (hFOB) cells, but the cells are never mentioned again in the manuscript.
Yes, but they are mentioned only in introduction section as a state of art. The article is not discussing the usage of the following cell line.
Lines 123-124: This sentence seems kind of vague as it is not clear exactly what is meant by the group's “successful fabrication” of a tissue engineering scaffold. Or is it just important that they fabricated a scaffold with a high amount of surface roughness in the nanometer range using an electro spraying technique? Ramier et al. showed that incorporation of nHA within/on their PHB fibers caused changes in their mechanical properties and cell response.
And
Line 124: Poly(3-hydroxybutyrate) is referred to only by its abbreviation PHB, but it is not a common enough polymer to give the abbreviation without further explanation.
Re-written to: Ramier et al. fabricated the tissue engineering scaffold with high surface roughness by electrospraying nHA on the Poly(3-hydroxybutyrate) (PHB) electrospun fibers [48].
Lines 126-127: This sentence does not make sense. It is unclear what the point the authors are trying to convey.
Re-written to: Garcia Garcia et al. reported the fabrication of a 3D multilayer scaffold with HA particles electrosprayed on the fibers.
Line 127: “Apatites” should not be plural here.
Re-written
Line 136: “Hydroxyapatite nanoparticles” should just be referred to as nHA. Depending on the definition of nHA.
The same as in Line 30 and 52: We have chosen nanohydroxyapatite to be used along the whole article text for consistency.
Lines 137-138: “Nanoparticles of hydroxyapatite” should just be referred to as nHA. Depending on the definition of nHA.
We have chosen nanohydroxyapatite to be used along the whole article text for consistency.
Line 142: “MHM” is used instead of “MHS”.
Re-written
Lines 147 and 149: The word “atomisation” is used, but the correct word is “atomization”.
Re-written
Line 151: The word properties should be removed form the end of the sentence.
Re-written
Line 151-153: Authors should not be referenced by number in this manner, keep it consistent with other portions of the manuscript like this within the text.
Line 156: “MHH” is used instead of “MHS”
Re-written
Lines 158-159: This sentence is a big vague and worded in a confusing manner.
Re-written
Lines 173-174: The sentence should read “…substrates are heated in a pressure vessel…” instead of “hearted”.
Re-written
Line 224: 0,5 w/v% should be written as 0.5 w/v% to maintain consistency throughout the manuscript.
Re-written
Line 304: The abbreviation “Mw” was defined on line 294.
The additional abbreviation was deleted.
Line 312: The full name for the abbreviation ICP-OES is never given.
The abbreviation was added.
Line 337: “Nominal stretch” should be called strain. Which is how it is defined in Figure 12 A.
Line 334: The word “deviding” should be spelled “dividing”.
Re-written
Line 375: SEM was previously defined so it should be used here, also the beginning of this sentence could be rewritten as "Analysis of SEM micrographs revealed..."
Re-written to:
Analysis of SEM micrographs revealed that fibers produced from PLGA, PDLLA and PDLLA/PLGA, have a regular texture and smooth surface (Figures 3A-F). Histograms with the diameters frequency show that the fiber’s thickness is in the range of 2 μm – 3 μm.
Line 432: No parentheses around TG and DTG abbreviation.
In our opinion it should be written as it is currently.
Line 442: 123,1° ± 2,1° should be 123.1° ± 2.4° to maintain consistency within the manuscript.
Re-written
Line 443: 106,9° ± 2,4° should be 106.9° ± 2.4° to maintain consistency within the manuscript.
Re-written
Line 461: Should change "fibres" to "fibers" for consistency.
All changed to “fibres” for consistency.
Line 475: The reference to Figure 11 is only given with respect to PLGA, but the figure has much more information than just those samples.
Re-written
Line 495: The word “types” should not be plural here.
Re-written
Line 496: Instead of referring samples as “ultrasonically coated” they should be referred to as “sonocoated” for consistency.
Re-written
Lines 525-526: Magnification should not be abbreviated in line 526, you must maintain consistency.
In our opinion the information about magnification should be present under this image to clearly identify the images.
Lines 535-537: This is not a complete sentence.
We don’t understand this comment. In our opinion it is complete.
Lines 558-560: Not all lines that have been changed since the last review have been highlighted, this is the one of those.
The last lines of the paragraph were changed, respectively:
Thus, the fibers coating made from highly biocompatible nanohydroxyapatite produced using the microwave MHS method used in the present study provided an environment potentially friendly for bone cells.
Line 572: This should be “weakly” instead of “weekly”.
Re-written
Line 589: Remain consistent with the nomenclature that has been used in this paper previously, namely that these are called nHA particles.
Word “particles” was added.
Lines 592-594: This sentence does not really make sense.
Yes, there was mistake and sentence was re-written.
Line 595: “Form” should be changed to “forming”.
Line 602: “Sono-deposited” should be changed to sonocoated since that is what they have been called up until this point.
Re-written
Lines 614-617: The word “the” in “…sonocoating could potentially improve the bone cells adhesion” does not need to be there. This sentence needs some work.
Re-written to:
This study provided the first evidence that nanohydroxyapatite deposited by means of sonocoating could potentially improve bone cells adhesion, structural stability, harmful pH variations, wetting properties and extend the degradation time of a membrane for possible application in treatment of periodontal disease.
Line 626: The short sentence here does not really contribute to the conclusion of the manuscript.
Re-written
Sincerely Yours,
Julia Higuchi
Corresponding Author

This manuscript is a resubmission of an earlier submission. The following is a list of the peer review reports and author responses from that submission.
Round 1
Reviewer 1 Report
Hybrid nanofibrous scaffolds made of a biopolyester and calium-phosphate mineral are promising materials in the field of bone regeneration applications. In this article, the authors present an original method (sonocoating) allowing an efficient coating of PLLA/PLGA electrospun fibers with nano-hydroxyapatite (nHA). The authors compared this novel coating strategy with the coating by electrospraying. The experimental data clearly support the conclusion; the article is well written and proposes a very clear and comprehensive study. In conclusion, I recommend the publication of this article in the journal Nanomaterials once the following minor points will be addressed:
1- Lines 38-43: In my opinion, the first paragraph of the introduction is out of scope of the article. It is more relative to a very general introduction of a textbook in nanotechnology.
2- Line 99: problem of reference.
3- Line 106: "It was reported by Zou et al. [32], that that" …
4- Instead of the very general well-known introduction a more suitable and dedicated introduction in the field covering the specific aspect of the article must be done. Indeed, various techniques were developed. Among them:
- Electrospinning of PLGA+HA:
Lao et al J. Mater. Sci.: Mater. Med. 2011, 22 (8), 1873−1884.
- Mixing gelatin dissolved in TFE with CaCl2 and Na2HPO4 prior to electrospinning to produce bone-like apatite after immersion in SBF:
Choi et al. Int. J. Biol. Macromol 2012; 50:1188-94.
- Electrospinning polyester + electrospraying HA. Ex:
Gupta et al. Biomaterials, 2009, 30(11), pp. 2085-2094
Francis et al. Acta Biomaterialia, 2010, 6(10), pp. 4100-4109
Ramier et al. Materials Science and Engineering C, 2014, 38(1), pp. 161-169
- Electrospinning PCL + electrosprayed nHA with patterned collectors to mimic the osteon structure:
Garcia Garcia et al., ACS Biomater. Sci. Eng. 2018; vol. 4, 3317–3326.
- Electrospun PCL-PEG-PCL containing 30 wt.% nHA
Fu et al. Biomaterials 2012; 33:8363-71.
- preparing coaxial polyester/gelatin electrospun nanofibers prior to mineralization the surface of the fibers (this technique allows also a homogeneous coverage by HA):
Cai et al. Mater. Lett 2013; 91:275-78.
Lazarini Pereira et al J. Mater. Sci.: Mater. Med., vol. 25, pp. 1137-1148, 2014.
Li X et al Langmuir 2008; 24:14145-50.
5- Legends of Fig11 and Fig12: To avoid confusion, please add the terms dry and wet in the legends of Fig11 and Fig12 respectively.
6- Lines 413-414. I don't understand the comment because the SEM picture of Fig 12D seems to show that the HA coated layer is not continuous and that the surface is only partially covered by HA. Can the authors add a supplementary picture at higher magnification and/or carry out FTIR - XPS characterizations after the tensile test?
Author Response
REVIEWER #1
Dear Reviewer,
We hereby submit the revised version of the manuscript initially entitled “Electrosprayed and Sonocoated with Nano-Hydroxyapatite Fibrous Biodegradable Polymer Membranes for Periodontal Tissues Regeneration” for consideration to be published in Nanomaterials Journal (Special Issue: Nanomaterials and Nanotechnology in Dentistry).
We appreciate all comments and suggestions. We have addressed them below raised point by point. The responses are provided and written in blue, whereas the changes introduced in the manuscript are copied here, highlighted in yellow.
Comments and Suggestions for Authors
Hybrid nanofibrous scaffolds made of a biopolyester and calium-phosphate mineral are promising materials in the field of bone regeneration applications. In this article, the authors present an original method (sonocoating) allowing an efficient coating of PLLA/PLGA electrospun fibers with nano-hydroxyapatite (nHA). The authors compared this novel coating strategy with the coating by electrospraying. The experimental data clearly support the conclusion; the article is well written and proposes a very clear and comprehensive study. In conclusion, I recommend the publication of this article in the journal Nanomaterials once the following minor points will be addressed:
Lines 38-43: In my opinion, the first paragraph of the introduction is out of scope of the article. It is more relative to a very general introduction of a textbook in nanotechnology.
The following sentences have been re-written according to the Reviewer suggestions to simplify the Authors’ introduction to the field of nanotechnology and its medical applications:
Emerging technologies and expanding knowledge about nanomaterials led in recent years to transformation of medical and dental clinical strategies. Major advances and innovations were made especially in the fields of nanomaterials for periodontal regeneration in order to replace damaged tissues[1,2,3].
2- Line 99: the problem of reference.
Fault reference was changed.
3-Line 106: "It was reported by Zou et al. [32], that that" …
Repetition was removed.
Instead of the very general well-known introduction a more suitable and dedicated introduction in the field covering the specific aspect of the article must be done. Indeed, various techniques were developed. Following the instruction from the Reviewer the introduction section with reference to currently known methods of nHA incorporation to composite electrospun materials was re-written:It was reported by Lao et al. that hydroxyapatite addition to PLGA electrospinning solution increased the proliferation of mouse pre-osteoblasts (MC3T3-E1) cultured on material [44]. Another known approach is the surface deposition of hydroxyapatite by electrospraying on the surface of materials [45]. Gupta et al. reported that electrospraying of HA nanoparticles in combination with electrospinning of fibers offered favorable surface topography and osteophilic environment for the attachment and growth of human fetal osteoblastic hFOB cells [46]. Ramier et al. successfully fabricated the tissue engineering scaffold by electrospraying nHA on the PHB electrospun fibers. Thus, creating a biocompatible scaffold for bone tissue regeneration with high surface roughness efficient for direct cells-bioceramics interactions [47]. Garcia Garcia et al. reported the fabrication of 3D multilayer scaffold with HA particles electrosprayed on the fibers which mimic the osteon structure in the bone. As a result of in vitro study for such structure, mesenchymal stem cells exhibited a preference for differentiation toward bone lineage without the presence of any other factors [48]. Another often suggested the method of apatites deposition on the materials is immersion in simulated body fluid (SBF) [49,50]. Apatite coatings fabricated by this method are precipitated in various topographies, such as flower-like, plate-like, needle-like structures of high brittleness [51,52]. Nevertheless, structures formed by this method are brittle and often delaminating when submitted to loading [53]. In our previous work, we proved that another promising method is sonocoating, which can be applied to fabricate the nanometric thickness HA coatings on polymeric scaffolds. The presence of sono-deposited nHA led to major bone tissue growth enhancement in 3 month-long in vivo test on New Zealand male rabbits [54].
Following References were added to the state of art (Introduction) section:
Lao, L.; Wang, Y.; Zhu, Y.; Zhang, Y.; Gao, C. Poly(lactide-co-glycolide)/hydroxyapatite nanofibrous scaffolds. Journal of Materials Science: Materials in Medicine, 2011, 22 (8), Pages: 1873−1884. – Referring to electrospinning of PLGA with the addition of hydroxyapatite Francis, L.; Venugopal, J.; Prabhakaran, M. P.; Thavasi, V.; Marsano, E.; Ramakrishna, S. Simultaneous electrospin–electrosprayed biocomposite nanofibrous scaffolds for bone tissue regeneration. Acta Biomaterialia,2010, 6(10), Pages: 4100–4109. Gupta, D.; Venugopal, J.; Mitra, S.; Giri Dev, V. R.; Ramakrishna, S. Nanostructured biocomposite substrates by electrospinning and electrospraying for the mineralization of osteoblasts. Biomaterials, 2009, 30(11), Pages: 2085–2094. Ramier, J.; Bouderlique, T.; Stoilova, O.; Manolova, N.; Rashkov, I.; Langlois, V.; Renard, E.; Albanese, P.; Grande, D.; Biocomposite scaffolds based on electrospun poly(3-hydroxybutyrate) nanofibers and electrosprayed hydroxyapatite nanoparticles for bone tissue engineering applications. Materials Science & Engineering C-Materials for Biological Applications, 2014, 38(1), Pages: 161-169. - Referring to the methods of electrospinning and electrospraying fibers with nanohydroxyapatite for bone tissue regeneration Garcia Garcia, A.; Hébraud, A.; Duval, J.L.; Wittmer, C.R.; Gaut, L.; Duprez, D.; Egles, C.; Bedoui, F.; Schlatter, G.; Legallais, C. Poly(ε-caprolactone)/Hydroxyapatite 3D Honeycomb Scaffolds for a Cellular Microenvironment Adapted to Maxillofacial Bone Reconstruction. ACS Biomaterials Science & Engineering, 2018, 49, Pages: 3317-3326. - Referring to the methods of electrospinning and electrospraying with hydroxyapatite to fabricate complex 3D structures for bone tissue regeneration. Cai, Q.; Feng, Q.; Liu, H; Yang X. Preparation of biomimetic hydroxyapatite by biomineralization and calcination using poly(l-lactide)/gelatin composite fibrous mat as template. Materials Letters, 2013, 91, 15, Pages: 275-278. Li, X.; Xie, J.; Yuan, X.; Xia, Y. Coating Electrospun Poly(ε-caprolactone) Fibers with Gelatin and Calcium Phosphate and Their Use as Biomimetic Scaffolds for Bone Tissue Engineering. Langmuir, 2008, 24, Pages:14145-14150. Fu, Q.W.; Zi, Y.P.; Xu W.; Zhou, R.; Cai, Z.Y.; Zheng, W.J.; Chen, F.; Qian, Q.R. Electrospinning of calcium phosphate-poly(D,L-lactic acid) nanofibers for sustained release of water-soluble drug and fast mineralization. International Journal of Nanomedicine, 2016, 11, Pages: 5087-5097. Li, L.; Li, G.; Jiang, J.; Liu, X.; Luo, L.; Nan, K. Electrospun fibrous scaffold of hydroxyapatite/poly (ε-caprolactone) for bone regeneration. Journal of Materials Science: Materials in Medicine, 2012, 2, Pages: 547-554. Xie, J.; Zhong, S.; Ma, B.; Shuler, F.D.; Lim, C.T. Controlled biomineralization of electrospun poly(ε-caprolactone) fibers to enhance their mechanical properties., Acta Biomaterialia, 2013, 9(3), Pages: 5698-5707. All, referring to the methods of electrospinning and biomimetic mineralization in simulated body fluid (SBF) of electrospun materials.
6- Lines 413-414. I don't understand the comment because the SEM picture of Fig 12D seems to show that the HA-coated layer is not continuous and that the surface is only partially covered by HA. Can the authors add a supplementary picture at higher magnification and/or carry out FTIR - XPS characterizations after the tensile test?
We agree with the reviewer that the choice of words to describe the outcome of the stretching samples in PBS conditions was not accurate. Thus, we have re-written the following sentence for better understanding:
“The nHA layer produced using sonocoating did not crumble during deformation and even increased the elasticity during wet conditions tensile test. The nanoparticles after the test were still attached to the fibers but not forming a continuous layer.”
Base on our observations coatings when the material is stretched do not fall apart and behave differently from brittle ceramics. Probably due to nanometric thicknesses of the coating.

Reviewer 2 Report
Overall, this manuscript describes work that is of general interest to the readers of this journal, but there are issues with the study design that weaken the support for the authors’ claims. The relevance of the two coating methods that were compared is questionable since one is a non-continuous homogeneous mixture of GoHAP3 and GoHAP6 (electrospray), while the other is a more continuous heterogenous bi-layer system with distinct GoHAP3 and GoHAP6 layers (sonocoating). Furthermore, some statements are made about the enhanced or increased osteoconductivity, bioactivity, and osteopromotive capabilities of the fabricated fibrous scaffolds when these aspects were not directly tested in the presented work. Additionally, there were grammatical and English errors throughout the manuscript that need to be addressed. Detailed lists of major and minor revisions are detailed below. Major revisions: 1) In the electrosprayed samples there is a supposedly homogeneous distribution of the GoHAP3 and GoHAP6, but in the sonocoated samples there are two distinct layers with the GoHAP6 deposited first followed by a different layer of the GoHAP3. I do not think these two systems are comparable because one is a homogeneous mixture and one is heterogeneous. 2) The electrosprayed samples still had regions of exposed polymer, so the increased release of Ca2+ from sonocoated samples could be attributed to the higher amount of nHA that was present on those samples. 3) The two techniques used to coat the fibers do not generate similar coatings, with one being almost continuous and the other applying aggregates to the fiber surfaces. 4) The authors need to indicate the amount of nHA loaded into each sample type. 5) Some statements are too broad and nonspecific. In line 100, it is stated that “Moreover, when polymeric implants persist in the cavity it can often cause the release of acidic by products…”. This is only true of specific polymers that have acidic degradation products. 6) It is stated in line 121 that “To improve osteoconductivity and bioactivity of the electrospun polymer materials in this study two methods were proposed:…”, but the osteoconductivity and bioactivity of these samples were not evaluated in this study. Also in line 440 and 506, osteopromotive capacity and enhanced bone regeneration potential are mentioned, respectively, but not tested in this manuscript. 7) For some tests only one type of nHA coating was used, such as in Fig. 5 and 7, and it is unclear in Fig. 5 if the nHA coated PDLLA/PLGA samples were generated using the electrospray coating or sonocoating technique. 8) In line 352 it is stated that “The ultrasonic coating enabled the conversion of the scaffold into hydrophilic state and increased water spreading which could possibly enhance cellular attachment, proliferation, differentiation and growth on the coated membrane.” This is speculation, not a result. 9) There are no control fiber samples for the Ca2+ release or pH studies. 10) In line 384 it is claimed that the electrospinning process did not significantly change the molecular weight of either polymer. There is no support for this claim. Minor revisions: 1) Consistency needs to be ensured throughout the manuscript, such as use of “fiber” vs. “fibre” was inconsistent. 2) The FDA does approves devices, not specific polymers. This is a common misunderstanding. 3) In line 97 it is stated that “…PDLLA and PLGA mimic those of host tissues.” It is not entirely clear what aspects of the tissue that the authors are trying to say that these polymers mimic, but at a fundamental level these polymers do not mimic native tissue. 4) Errors with references need to be fixed, such as in line 99. 5) Reference 33 does not talk specifically about neutralization of acidic environment due to water absorption, which is what it is being referenced for. 6) There are parts of the manuscript that appear in the wrong sections, such as line 142 of the Materials section it is stated that “They [the GoHAP3 and GoHAP6] were successfully applied for coating of bone regrowth scaffolds using sonocoating technology”. 7) In Fig. 9 it is not clear exactly what the viewer is looking at, these images need to be labeled for ease of reading. 8) It is stated in line 388 that samples “… lost physical integrity after 8 weeks…” it needs to be more clear exactly what it means to lose physical integrity, this is too vague. 9) Inconsistencies between figure captions and text need to be addressed, i.e. line 403 states that Fig. 11 is for dry mechanical testing, but in the figure caption it states these samples were tested in PBS. 10) The issue of scalability in electrospraying is brought up, but the scalability of the sonocoating technique is never addressed. 11) In line 476 it is stated that “The resorption of hydroxyapatite layer of polymer fibers observed in this study…”, but no resorption occurred here, possibly the wrong word was used.Author Response
Reviewer #2
Dear Reviewer,
We hereby submit the revised version of the manuscript initially entitled “Electrosprayed and Sonocoated with Nano-Hydroxyapatite Fibrous Biodegradable Polymer Membranes for Periodontal Tissues Regeneration” for consideration to be published in Nanomaterials Journal (Special Issue: Nanomaterials and Nanotechnology in Dentistry).
We appreciate all comments and suggestions. We have addressed them below raised point by point. The responses are provided and written in blue, whereas the changes introduced in the manuscript are copied here, highlighted in yellow.
Comments and Suggestions for Authors:
Overall, this manuscript describes work that is of general interest to the readers of this journal, but there are issues with the study design that weaken the support for the authors’ claims. The relevance of the two coating methods that were compared is questionable since one is a non-continuous homogeneous mixture of GoHAP3 and GoHAP6 (electrospray), while the other is a more continuous heterogenous bi-layer system with distinct GoHAP3 and GoHAP6 layers (sonocoating).
We agree with the Reviewer that coatings obtained by electrospraying and sonocoating methods possess different structural characteristics. However, in this work, a cost-efficient alternative method (sonocoating) to the traditional techniques, such as (electrospraying) or biomimetic mineralization to produce a hydroxyapatite (HA) coating with a nanostructured feature onto a fibers surface is presented. Our intention was to show the prospective of a novel sonocoating method in comparison with well-known elecrospraying technoloqy.
We are aware that there are many comparative methods providing continuous coating layer (SBF immersion, plasma spraying, magnetron sputtering, sol–gel or pulsed laser deposition). However, these methods involve extreme high temperature or are high cost or simply technically cumbersome. Nevertheless, electrospraying was chosen for comparison because similarly to sonocoating, deposition of HA nanocrystals onto fibrous substrates can be performed at room temperature, using nHA suspensions as the starting solution and can be successfully used without temperature-related change in nHA structure. Low-temperature deposition is also important for preserving the thermodynamically unstable nanostructure of HAP particles, which may increase grain size in higher temperatures and lost their unique biological properties. Biomimetic mineralization by SBF immersion for example provide continuous coating but does not provide the accurate nanohydroxyapatite structure and it is not applicable when biomaterial big-scale manufacturing is considered.
Furthermore, some statements are made about the enhanced or increased osteoconductivity, bioactivity, and osteopromotive capabilities of the fabricated fibrous scaffolds when these aspects were not directly tested in the presented work.
Following the instruction from the Reviewer, we have added the in vitro culture study results of the samples against the MG-63 osteosarcoma cells and BJ fibroblasts cells to support our claim. The rationale for using the following cell lines was the suggested future application of the material – barrier membrane between bone and soft tissues. Also, MG-63 and BJ cell lines have been selected for the study due to wide use and well-defined protocols for bone regeneration biomaterials testing.
In the electrosprayed samples there is a supposedly homogeneous distribution of the GoHAP3 and GoHAP6, but in the sonocoated samples there are two distinct layers with the GoHAP6 deposited first followed by a different layer of the GoHAP3. I do not think these two systems are comparable because one is a homogeneous mixture and one is heterogeneous.
Indeed, two types of coatings differ. Nevertheless, Authors of this study performed a painstaking literature review and concluded that electrospraying is the most comparative method in terms of as-synthesized nHA deposition on electrospun fiber without causing the structural changes to the thermodynamically unstable nanostructure of HAP particles, which again may increase grain size and loose unique biological properties when used in other deposition techniques. There are also other low-temperature deposition methods, such as biomimetic mineralization, but the final coating material obtained in such a manner does not provide the features of the nHA particles synthesized in microwave solvothermal synthesis.
2) The electrosprayed samples still had regions of exposed polymer, so the increased release of Ca2+ from sonocoated samples could be attributed to the higher amount of nHA that was present on those samples.
We would like to thank the Reviewer for the good suggestion. To enable a better understanding of the coating behavior on the fibers we have performed the Thermogravimetric Analysis to measure the content of the HA nanoparticles on the electrospun fibers. Thus we added the following information about the measurement and composition of each sample type.
To the “2.5. Physico-chemical characterization of materials” sub-section:
“ The content of the HA nanoparticles on the electrospun fibers was measured using thermogravimetric analysis (TGA). To record the change in sample mass during heating samples were measured on a thermogravimetric analyzer (STA 449 F1 Jupiter®, Netzsch). The samples were heated from 20 to 600 °C at a rate of 10 °C/min under a helium (He) atmosphere at a flowrate of 60 mL min−1. Approximately 10 mg of sample was heated in alumina crucible. “
To the “3.4. Thermogravimetric analysis (TGA)”sub-section:
“ The thermogravimetric analysis of the membranes showed differences in the thermal stability between tested samples. TGA (Figure 8A) and DTG (Figure 8B) results showed that there was a increase in melting temperature of the material from 360oC to 395 oC from non-modified PDLLA/PLGA to sonocoated PDLLA/PLGA with the highest content of nHA. The residual weight for nHA modified samples was 3,6 wt% for electrosprayed samples and 13,6% for sonocoated, respectively.
As a result of the Thermogravimetric Analysis the residual weight of samples indicate that the amount of nHA on the surface of the sonocoated samples was indeed higher than electrosprayed.
3) The two techniques used to coat the fibers do not generate similar coatings, with one being almost continuous and the other applying aggregates to the fiber surfaces.
Again, as in comment nr. 1:
Indeed, two types of coatings differ. Nevertheless, Authors of this study performed painstaking literature review and concluded that electrospraying is the most comparative method in terms of as-synthetized nHA deposition on electrospun fiber without causing the structural changes to the thermodynamically unstable nanostructure of HAP particles, which again may increase grain size and loose unique biological properties when used in other deposition techniques. There are also other low-temperature deposition methods, such as biomimetic mineralization, but the final coating material obtained in such a manner does not provide the features of the nHA particles synthetized in microwave solvothermal synthesis. Many studies have shown that the bioactivity of the hydroxyapatite is dependent on the size and morphology of the particles. Our nanoparticles used for the study already exhibit high bioactivity. Thus, it was our intention to apply those nanoparticles by method patented by Authors of this study and another chosen method which enables the usage of the same nanoparticles. These nanoparticles were already successfully applied in sonocoating of 3D printed materials and underwent extensive in vitro and in vivo implantation studies (including painstaking process of immunohistochemical analysis of the explanted tissues).
4) The authors need to indicate the amount of nHA loaded into each sample type.
The nHA content was measured by Thermogravimetric Analysis as answered in question 2.
5) Some statements are too broad and nonspecific. In line 100, it is stated that “Moreover, when polymeric implants persist in the cavity it can often cause the release of acidic by products…”. This is only true of specific polymers that have acidic degradation products.
We did specified this line as follows:
“The most importantly, chosen materials such as PDLLA and PLGA mimic those of host native tissues. Nevertheless, PDLLA-based materials have a slow degradation rate so that the degradation period may be up to 12 months due to the hydrophobic nature, which limits their use in GTR/GBR [24,25,26]. In order to regulate the degradation rate and hydrophilicity of PDLLA, we suggested blending with faster degrading PLGA [27,28,29,30,31]. However, the major limitation in use of PLGA is the acidity of the degradation products. When released in large quantities, can jeopardize efficient metabolization by the body causing an immune response [32]. To neutralize the degradation process, electrospun fibers can be coated with nHA which is chemically comparable to the mineral phase found in natural bone and it has been widely used in bone tissue engineering due to its bioactivity and osteoconductivity. It was reported by Zou et al. [33], that that presence of hydroxyapatite in polymeric matrices led to high water absorption, caused neutralization of the acidic environment [34] and slowed down autocatalysis during polymer biodegradation [35].”
6) It is stated in line 121 that “To improve osteoconductivity and bioactivity of the electrospun polymer materials in this study two methods were proposed:…”, but the osteoconductivity and bioactivity of these samples were not evaluated in this study. Also in line 440 and 506, osteopromotive capacity and enhanced bone regeneration potential are mentioned, respectively, but not tested in this manuscript.
We agree with the Reviewer that the effect of bone growth enhancement attributed to the presented coatings is currently speculative. Nevertheless, the previous studies of the nanoparticles used for the study (in vitro, in vivo studies) indicate that the positive influence of the presence of the coating in obvious. The most important result of the previous studies presented by Rogowska-Tylman et al. was that in comparing the 3D-printed PCL coated with exact nanoparticles and uncoated scaffolds, we found a striking difference of tissue responses between them. For the uncoated PCL scaffold, only an insignificant amount of bone tissue, occasionally on the surface of the polymer fibers was found. This suggests the effect of the coating itself. To better understand this effect of coating presence we performed in vitro tests against osteoblastic and fibroblastic cells as written in previous answers. Moreover, we agree with the Reviewer that some sentences in the manuscript text are too speculative. Thus we have changed the following parts of the conclusions section:
7) For some tests only one type of nHA coating was used, such as in Fig. 5 and 7, and it is unclear in Fig. 5 if the nHA coated PDLLA/PLGA samples were generated using the electrospray coating or sonocoating technique. 8) In line 352 it is stated that “The ultrasonic coating enabled the conversion of the scaffold into hydrophilic state and increased water spreading which could possibly enhance cellular attachment, proliferation, differentiation and growth on the coated membrane.” This is speculation, not a result.
We agree with the Reviewer that the sentence was wrongly placed in Results section. Thus, the sentence was moved to the conclusions section.
9) There are no control fiber samples for the Ca2+ release or pH studies.
This data was added to the pH graph according to the Reviewer comment. Nevertheless, in our opinion there is no point of adding line corresponding to the polymer to the graph of Ca2+ ions release because there were no present in the sample.
10) In line 384 it is claimed that the electrospinning process did not significantly change the molecular weight of either polymer. There is no support for this claim.
The results of the gel-permeation chromatography analysis have shown that the molecular weight for the materials after electrospinning were the same: in the ranges given for pristine polymers. PLGA Mw=100-130kDa, PDLLA Mw=200-220kDa, PDLLA/PLGA Mw=150-180kDa.
Minor revisions: 1) Consistency needs to be ensured throughout the manuscript, such as use of “fiber” vs. “fibre” was inconsistent.
Changed according to comment to obtain concise vocabulary.
The FDA does approves devices, not specific polymers. This is a common misunderstanding.
Thank you for this comment.
In line 97 it is stated that “…PDLLA and PLGA mimic those of host tissues.” It is not entirely clear what aspects of the tissue that the authors are trying to say that these polymers mimic, but at a fundamental level these polymers do not mimic native tissue.
We did specify our way of thinking.
4) Errors with references need to be fixed, such as in line 99. Done.
5) Reference 33 does not talk specifically about neutralization of acidic environment due to water absorption, which is what it is being referenced for.
We have changed the reference according to suggestion.
6) There are parts of the manuscript that appear in the wrong sections, such as line 142 of the Materials section it is stated that “They [the GoHAP3 and GoHAP6] were successfully applied for coating of bone regrowth scaffolds using sonocoating technology”.
We have changed the location of this section.
7) In Fig. 9 it is not clear exactly what the viewer is looking at, these images need to be labeled for ease of reading.
We have added the names over the images to provide better visibility.
8) It is stated in line 388 that samples “… lost physical integrity after 8 weeks…” it needs to be more clear exactly what it means to lose physical integrity, this is too vague.
Sample simply fell apart to pieces after 8 weeks of immersion
9) The issue of scalability in electrospraying is brought up, but the scalability of the sonocoating technique is never addressed.
We have included the additional information about the already present industrial sonocoating with the reference to the company which is providing such services.
11) In line 476 it is stated that “The resorption of hydroxyapatite layer of polymer fibers observed in this study…”, but no resorption occurred here, possibly the wrong word was used.
Thank you for this comment. It was a mistake in vocabulary indeed. We have changed this sentence.

Reviewer 3 Report
This study presents an interesting/innovative approach for membranes aimed to tissue/bone regeneration, being supported by previous work of the same research team concerning the development of polymeric fibrous membranes containing surface nHA coatings (Refs 46, 47, 48, 50). The developed membranes were produced and characterized by conventional methodology.
Membranes’ degradation was analysed in detail and the underlying mechanisms were discussed taking into account wettability, polymer/nHA dissolution, morphological features of the degraded membranes, … Based on the conducted mechanical tests, wettability and degradation studies, authors concluded that “membranes are promising biocompatible materials for regeneration of cavities in periodontal tissue”. Also, in the beginning of the Discussion section, they stated that “The appropriate design of the membranes proposed in this study was proved to be promising for osteopromotive capacity enhancement of the materials”. However, the short (cell adhesion and spreading) and long-term (proliferation and differentiation) behavior of periodontal/bone cells on these membranes were not addressed. At least, appropriate in vitro cytocompatibility studies should be performed to support the suggested bone/periodontal regenerative application. This can not be inferred from published studies addressing other different HA-coated membranes. Even in the present study, differences in the cytocompatibility are expected between electrosprayed and sonocoated membranes. The initial (around day 1) high ionic release (Ca2+) observed on the nHA-coated sonocoated membranes is expected to affect cell adhesion and spreading compared to that on the electrosprayed membrane (not presenting a burst ion release), i.e. the initial localized high ionic strength might hinder cell adhesion to the sonocoated membrane.
In the same line, the title “Electrosprayed and Sonocoated with Nano-Hydroxyapatite Fibrous Biodegradable Polymer Membranes for Periodontal Tissues Regeneration” is not supported by the results. Authors should be more conservative – the Title should reflect the reported results, and not be speculative.
Author Response
REVIEWER #2
Dear Reviewer,
We hereby submit the revised version of the manuscript initially entitled “Electrosprayed and Sonocoated with Nano-Hydroxyapatite Fibrous Biodegradable Polymer Membranes for Periodontal Tissues Regeneration” for consideration to be published in Nanomaterials Journal (Special Issue: Nanomaterials and Nanotechnology in Dentistry).
We appreciate all comments and suggestions. We have addressed them below raised point by point. The responses are provided and written in blue, whereas the changes introduced in the manuscript are copied here, highlighted in yellow.
This study presents an interesting/innovative approach for membranes aimed to tissue/bone regeneration, being supported by previous work of the same research team concerning the development of polymeric fibrous membranes containing surface nHA coatings (Refs 46, 47, 48, 50). The developed membranes were produced and characterized by conventional methodology.
Comments and Suggestions for Authors
Membranes’ degradation was analysed in detail and the underlying mechanisms were discussed taking into account wettability, polymer/nHA dissolution, morphological features of the degraded membranes, … Based on the conducted mechanical tests, wettability and degradation studies, authors concluded that “membranes are promising biocompatible materials for regeneration of cavities in periodontal tissue”. Also, in the beginning of the Discussion section, they stated that “The appropriate design of the membranes proposed in this study was proved to be promising for osteopromotive capacity enhancement of the materials”. However, the short (cell adhesion and spreading) and long-term (proliferation and differentiation) behavior of periodontal/bone cells on these membranes were not addressed. At least, appropriate in vitro cytocompatibility studies should be performed to support the suggested bone/periodontal regenerative application.
Following the instruction from the Reviewer, we have added the in vitro culture study results of the samples against the MG-63 osteosarcoma cells and BJ fibroblasts cells to support our claim. The rationale for using the following cell lines was the suggested future application of the material – barrier membrane between bone and soft tissues. Also, MG-63 and BJ cell lines have been selected for the study due to wide use and well-defined protocols for bone regeneration biomaterials testing.
Samples in vitro cytotoxicity to MG-63 and BJ cells
The cytotoxic activity of PDLLA/PLGA and PDLLA/PLGA/nHA sonocoated membranes were investigated on the human osteosarcoma cells MG-63 and BJ skin fibroblasts. Cells were seeded on 96-microwell plates with tested membranes, and the measurements were done 24 h, 72 h and seven days after seeding. The influence of coating on MTS assay was determined. As shown in the Figure B metabolic activity of MG-63 cells on the surface of PDLLA/PLGA, and PDLLA/PLGA/nHA sonocoated membranes were similar for all tested materials after first 24 h. However, after one week of cell culture, the metabolic activity varied for PDLLA/PLGA and PDLLA/PLGA/nHA sonocoated membranes. A significantly increased mitochondrial activity was observed mostly for PDLLA/PLGA/nHA sonocoated membranes after seven days of cell seeding. In contrast, seeding BJ cells (Figure … A) on PDLLA/PLGA/nHA sonocoated membranes causes an increase in cell proliferation, but it is not as large as with MG-63 cells.
Figure 13. Metabolic activity (MTS assay) of MG-63 (A) and BJ (D) cells seeded on PDLLA/PLGA and PDLLA/PLGA/nHA sonocoated materials after 24h, 72h and 7 days. SEM colorized images of MG-63 cells attachment after 24h on (B-C) PDLLA/PLGA and (E-F) PDLLA/PLGA/nHA sonocoated materials. Data are expressed as a percentage of the control, mean ± SD from three independent experiments. (*denotes statistically significant difference from unexposed control, p<0.05).
This cannot be inferred from published studies addressing other different HA-coated membranes. Even in the present study, differences in the cytocompatibility are expected between electrosprayed and sonocoated membranes. The initial (around day 1) high ionic release (Ca2+) observed on the nHA-coated sonocoated membranes is expected to affect cell adhesion and spreading compared to that on the electrosprayed membrane (not presenting a burst ion release), i.e. the initial localized high ionic strength might hinder cell adhesion to the sonocoated membrane.
The results of the performed tests (cells metabolic activity against samples) indicate that nHA sonocoated materials do not induce cytotoxic effect. Contrary, there was a constant growth in metabolic activity of MG-63 cells on the surface of PDLLA/PLGA/nHA sonocoated membranes. After 24h of culture results were similar for for all tested materials. However, after 7 days of cell culture, significantly increased mitochondrial activity was observed mostly for PDLLA/PLGA/nHA sonocoated membranes. In contrast, seeding BJ cells on PDLLA/PLGA/nHA sonocoated membranes causes an increase in cell proliferation, but it is not as large as with MG-63 cells. This indicate the favorable surface environment for bone cells growth and proliferation.
In the same line, the title “Electrosprayed and Sonocoated with Nano-Hydroxyapatite Fibrous Biodegradable Polymer Membranes for Periodontal Tissues Regeneration” is not supported by the results. Authors should be more conservative – the Title should reflect the reported results, and not be speculative.
We agree with the reviewer thus we have added cell culture results as stated above to support our manuscript title.

Round 2
Reviewer 2 Report
Major Revisions:
Points 4, 5, 8, and 9 of previous notes were adequately addressed.
Selected cell lines are in fact well established however, MG-63 osteosarcoma as well as BJ fibroblasts are not themselves indicative of an osteoconductive or osteopromotive environment. There was no assessment of the osteogenic nature of these fibrous scaffolds. Although authors claim that nHA coatings caused “major bone formation” in rabbit model [55] (lines 136-140), it appears that this is based upon the H&E stained sections that are presented in Fig. 16 of that work and morphometric analysis in Fig. 17. However, H&E is not a definitive stain for bone formation it is a general histological tissue stain. It feels like a low bar was set for determining the amount of bone formation.
1) I would like to see a bit more of this type of validation within the manuscript itself. Although my questions are answered in the comments they are not adequately answered in the body of the work itself.
2) The higher amount of nHA on the sonocoated samples is addressed however this data is never used to make any statements about the resultant increase in realeased Ca2+ or used for any sort of normalization between samples.
3) Again, while the concerns are being justified in the comments, this also needs to be reflected in the manuscript so that it is clear why the two techniques specifically are being compared to one another.
6) Although authors (Rogowska-Tylman et al.) claim that nHA coatings caused “major bone formation” in rabbit model, it appears that this is based upon the H&E stained sections that are presented in Fig. 16 of that work and morphometric analysis in Fig. 17. However, H&E is not a definitive stain for bone formation - it is a general histological tissue stain.
7) It should be demonstrated that the electrosprayed samples have similar chemical structure and elemental surface chemistry to sonocoated samples or at least stated that they were similar.
10) This may be true, but it is not shown. I would suggest either removing the mentioning of no change in molecular weight or cite a source that supports this claim.
11) There is no Figure 12 in the most recent version of the manuscript making it impossible to determine its accuracy and validity
12) The labels for parts A and B of figure 13 do not match what is written in the caption and body of the text, which makes the text confusing.
Minor Revisions:
Points 4, 5, 6, 7, 9, and 11 of previous notes were adequately addressed.
1) Issue with word fiber was resolved, but there is now an issue with the abbreviation of DTGA in the body of the work vs DTG in the caption.
2) This was not addressed in revisions.
3) Although this is following a sentence that talks about the “mechanical functionality” of electrospun membranes it is never specifically stated what is being mimicked. The sentence “The most importantly, chosen materials such as PDLLA and PLGA mimic those of host native tissues.” does not make sense.
8)This still remains an issue, it is still not stated what it meant for the membranes to “lose physical integrity”. That could mean anything from complete dissolution to simply not being able to support a load.
10) was not listed on returned notes from Authors.
Author Response
Dear Reviewer,
Please see the Answers in PDF file attached.
Yours sincerely,
Corresponding Author

Reviewer 3 Report
The authors improved the manuscript according to the suggestions.
Author Response
Dear Reviewer,
Thank you for your comments and suggestions. They were crucial for the improvement of the following manuscript text.
Yours Sincerely,
Corresponding Author